# ESCRT-I fuels lysosomal degradation to restrict TFEB/TFE3 signaling via the Rag-mTORC1 pathway

Marta Wróbel[1],* , Jarosław Cendrowski[1],* , Ewelina Szymańska[1] , Malwina Grębowicz-Maciukiewicz[1] , Noga Budick-Harmelin[1], Matylda Macias[2], Aleksandra Szybińska[2], Michał Mazur[1], Krzysztof Kolmus[1] , Krzysztof Goryca[3], Michalina Dąbrowska[3], Agnieszka Paziewska[4], Michał Mikula[3], Marta Miączyńska[1]

Within the endolysosomal pathway in mammalian cells, ESCRT complexes facilitate degradation of proteins residing in endosomal membranes. Here, we show that mammalian ESCRT-I restricts the size of lysosomes and promotes degradation of proteins from lysosomal membranes, including MCOLN1, a Ca$^{2+}$ channel protein. The altered lysosome morphology upon ESCRT-I depletion coincided with elevated expression of genes annotated to biogenesis of lysosomes due to prolonged activation of TFEB/TFE3 transcription factors. Lack of ESCRT-I also induced transcription of cholesterol biosynthesis genes, in response to inefficient delivery of cholesterol from endolysosomal compartments. Among factors that could possibly activate TFEB/TFE3 signaling upon ESCRT-I deficiency, we excluded lysosomal cholesterol accumulation and Ca$^{2+}$-mediated dephosphorylation of TFEB/TFE3. However, we discovered that this activation occurs due to the inhibition of Rag GTPase–dependent mTORC1 pathway that specifically reduced phosphorylation of TFEB at S122. Constitutive activation of the Rag GTPase complex in cells lacking ESCRT-I restored S122 phosphorylation and prevented TFEB/TFE3 activation. Our results indicate that ESCRT-I deficiency evokes a homeostatic response to counteract lysosomal nutrient starvation, that is, improper supply of nutrients derived from lysosomal degradation.

## Introduction

Lysosomes are acidic organelles of animal cells that serve as a major degradative compartment for proteins or lipids delivered by endocytosis or autophagy (Luzio et al, 2007), hence are equivalent of a yeast vacuole. Besides providing cells with metabolites derived from degradation and with molecules taken up by endocytosis, lysosomes also act as signaling organelles (Ballabio, 2016). Reduced cargo delivery to lysosomes or their dysfunction induces lysosome-related signaling pathways that adjust cellular metabolism (Ballabio, 2016). These pathways are orchestrated by kinases activated from the lysosomal surface or by changes in efflux of metabolites or ions from the lysosomal lumen.

Key mediators of signaling activated from lysosomes are transcription factors belonging to the MiT-TFE family, such as TFEB and TFE3, that upon activation, translocate to the nucleus and induce transcription of target genes involved in biogenesis of lysosomes (Pena-Llopis et al, 2011; Martina et al, 2012; Roczniak-Ferguson et al, 2012; Settembre et al, 2012). When lysosomes are functional and nutrients are abundant, these factors remain inhibited because of their phosphorylation by mTORC1 (mechanistic target of rapamycin complex 1) kinase that prevents their nuclear translocation (Settembre et al, 2012; Vega-Rubin-de-Celis et al, 2017). mTORC1-dependent phosphorylation of MiT-TFE factors is mediated by Rag GTPases that recruit MiT-TFE proteins to lysosomes and promote mTORC1 kinase activity (Martina & Puertollano, 2013).

MiT-TFE nuclear translocation may be induced by a number of lysosome-related signaling cues. Reduced nutrient availability inhibits mTORC1-dependent phosphorylation of MiT-TFE factors by inactivation of Rag GTPases (Martina & Puertollano, 2013). Lysosomal dysfunction causes a release of calcium ions (Ca$^{2+}$) from lysosomes via channels formed by the mucolipin1 protein (MCOLN1, also known as TRPML1) which activates calcineurin, calcium-dependent phosphatase (Medina et al, 2015; Zhang et al, 2016). Calcineurin can dephosphorylate TFEB and TFE3, enabling their nuclear translocation (Medina et al, 2015; Martina et al, 2016;

[1]Laboratory of Cell Biology, International Institute of Molecular and Cell Biology, Warsaw, Poland   [2]Microscopy and Cytometry Facility, International Institute of Molecular and Cell Biology, Warsaw, Poland   [3]Department of Genetics, Maria Skłodowska-Curie National Research Institute of Oncology, Warsaw, Poland   [4]Department of Gastroenterology, Hepatology and Clinical Oncology, Medical Center for Postgraduate Education, Warsaw, Poland

Correspondence: miaczynska@iimcb.gov.pl; jcendrowski@iimcb.gov.pl
Noga Budick-Harmelin's present address is The Shmunis School of Biomedicine and Cancer Research, the George S. Wise Faculty of Life Sciences, Tel Aviv University, Tel Aviv, Israel.
Krzysztof Goryca's present address is Centre of New Technologies, Warsaw, Poland.
Agnieszka Paziewska's present address is Department of Neuroendocrinology, Centre of Postgraduate Medical Education, Warsaw, Poland and Faculty of Medical and Health Sciences, Institute of Health Sciences, Siedlce University of Natural Sciences and Humanities, Siedlce, Poland.
*Marta Wróbel and Jarosław Cendrowski contributed equally to this work.

Zhang et al, 2016). However, the involvement of Ca$^{2+}$ signaling in MiT-TFE regulation may be more complex as lysosomal Ca$^{2+}$ has also been shown to promote mTORC1 activity (Li et al, 2016) and therefore could potentially inhibit MiT-TFE signaling (Grimm et al, 2018).

Another example of adjusting cellular metabolism in response to altered lysosomal function is activation of transcription factors inducing the expression of genes responsible for cholesterol biosynthesis (Luo et al, 2020). It occurs in response to inefficient efflux of cholesterol from late endosomes or lysosomes that leads to its impaired delivery to the ER (Luo et al, 2020). Recently, abnormal cholesterol accumulation in lysosomes was shown to increase the cytosolic pool of Ca$^{2+}$ (Tiscione et al, 2019) and to promote nuclear accumulation of MiT-TFE factors (Willett et al, 2017).

Lysosomes are the main intracellular compartment where the turnover of membrane proteins occurs (Trivedi et al, 2020). Most of these proteins are delivered to lysosomes by means of endosomal trafficking via a sorting mechanism facilitated by endosomal sorting complexes required for transport (ESCRTs). ESCRTs encompass several protein assemblies (ESCRT-0, I, II, and III) that mediate membrane remodeling processes in endocytosis, autophagy, cytokinesis, nuclear envelope sealing, and virus budding (Vietri et al, 2020). During endosomal sorting, ESCRTs act sequentially to incorporate membrane proteins marked for degradation by ubiquitin into the lumen of endocytic organelles called late endosomes or multivesicular bodies (MVBs) (Raiborg & Stenmark, 2009; Wenzel et al, 2018). This occurs by invagination and scission of the endosomal outer (limiting) membrane, thereby forming intraluminal vesicles (ILVs). Upon fusion of late endosomes with lysosomes, intraluminal vesicles and their cargo reach the lysosomal lumen to be degraded. Inhibition of ESCRT-dependent cargo sorting on endosomes causes accumulation of membrane proteins, such as plasma membrane receptors, on the endosomal limiting membranes that may activate intracellular signaling pathways (Szymanska et al, 2018). As we previously showed, depletion of core ESCRT-I subunits, Tsg101 or Vps28, induces NF-κB signaling initiated by cytokine receptor clustering on endosomal structures (Maminska et al, 2016; Banach-Orlowska et al, 2018).

Recent reports uncovered a new role of ESCRT complexes, including ESCRT-I, in the turnover of proteins at yeast vacuolar membranes (Zhu et al, 2017; Morshed et al, 2020; Yang et al, 2020; Yang et al, 2021). Up to date, mammalian ESCRT proteins have been shown to associate with lysosomes only to restore their integrity disrupted by damaging agents (Radulovic et al, 2018; Skowyra et al, 2018; Jia et al, 2020). With the exception of one very recent report showing that some ESCRT-III components and Vps4 mediate degradation of several lysosomal membrane proteins (Zhang et al, 2021), the role of ESCRTs in the physiological regulation of lysosomal morphology, function, or signaling in the absence of induced lysosomal damage has not been thoroughly addressed. This question is important for human health as lysosomal function and signaling contribute to development or progression of cancer (Tang et al, 2020; Machado et al, 2021), including colorectal cancer (CRC), in which ESCRT machinery has been proposed as a promising therapeutic target (Szymanska et al, 2020; Kolmus et al, 2021).

Here, we show that the ESCRT-I complex mediates degradation of lysosomal membrane proteins, restricting lysosome size in human colorectal cancer (CRC) cells. Hence, ESCRT-I fuels lysosomes with cargo destined for degradation not only from endocytic and autophagic compartments but also from lysosomal membrane turnover. Therefore, ESCRT-I deficiency activates transcriptional responses, including Rag GTPase–dependent TFEB/TFE3 signaling, indicative of lysosomal nutrient starvation, that is, impaired nutrient delivery from lysosomes.

## Results

### ESCRT-I deficiency leads to appearance of enlarged structures positive for lysosomal markers

To address whether ESCRT-I is involved in lysosomal membrane homeostasis and signaling, we used CRC cells, in which we previously reported ESCRT-I to regulate cell growth and intracellular signaling (Kolmus et al, 2021). Using an siRNA-mediated approach, we efficiently depleted key ESCRT-I subunits, Tsg101 or Vps28, in RKO cells (Fig S1A). Of note, the removal of one of them strongly reduced the levels of the other because of the ESCRT-I complex destabilization, as described (Bache et al, 2004; Kolmus et al, 2021). First, we analyzed by confocal microscopy whether knockdown of Tsg101 or Vps28 in RKO cells affected the intracellular distribution of LAMP1, a marker of late endosomes and lysosomes (Fig S1B). We noticed that ESCRT-I deficiency increased the area of LAMP1-positive vesicular structures (Fig S1B), suggesting that late endosomes and/or lysosomes were enlarged.

To confirm the enlargement of lysosomes in the absence of ESCRT-I, we measured the effect of Tsg101 or Vps28 depletion on the intracellular distribution of LysoTracker, a cell-permeable fluorescent dye that accumulates in nondamaged acidic lysosomes (Pierzynska-Mach et al, 2014) (Fig 1A). Lack of ESCRT-I augmented the LysoTracker staining intensity per lysosome and increased the size of lysosomes (Fig 1B). Strong LysoTracker accumulation in lysosomes upon ESCRT-I deficiency indicated that their integrity was not impaired and that they maintained an acidic pH, important for their degradative function (Ballabio, 2016). To rule out the possibility that enlarged lysosomes in cells lacking ESCRT-I could be because of potential off-target effects of siRNAs, we silenced the expression of the gene encoding Tsg101 in RKO cells using the CRISPR/Cas9 approach (Fig S1C). As in the case of siRNAs, depletion of Tsg101 using two single-guide RNAs reduced the level of Vps28 protein (Fig S1C) and increased the average size of LysoTracker-positive structures (Fig S1D). To verify that the observed effects of ESCRT-I depletion on lysosomal size are not specific only to RKO cells, we depleted Tsg101 or Vps28 in another CRC cell line, DLD-1 (Fig S2A). Reassuringly, we observed increased LysoTracker staining intensity and size of lysosomes (Fig S2B–D).

To address whether ESCRT-I depletion affects the composition of lysosomes, we analyzed the intracellular distribution of active cathepsin D, a major lysosomal endopeptidase, by staining of fixed RKO cells with its inhibitor pepstatin A conjugated to BODIPY FL (Chen et al, 2000). We observed that cells lacking Tsg101 and Vps28 had increased levels of active cathepsin D, particularly in the lumen of enlarged LAMP1 structures (Fig 1C).

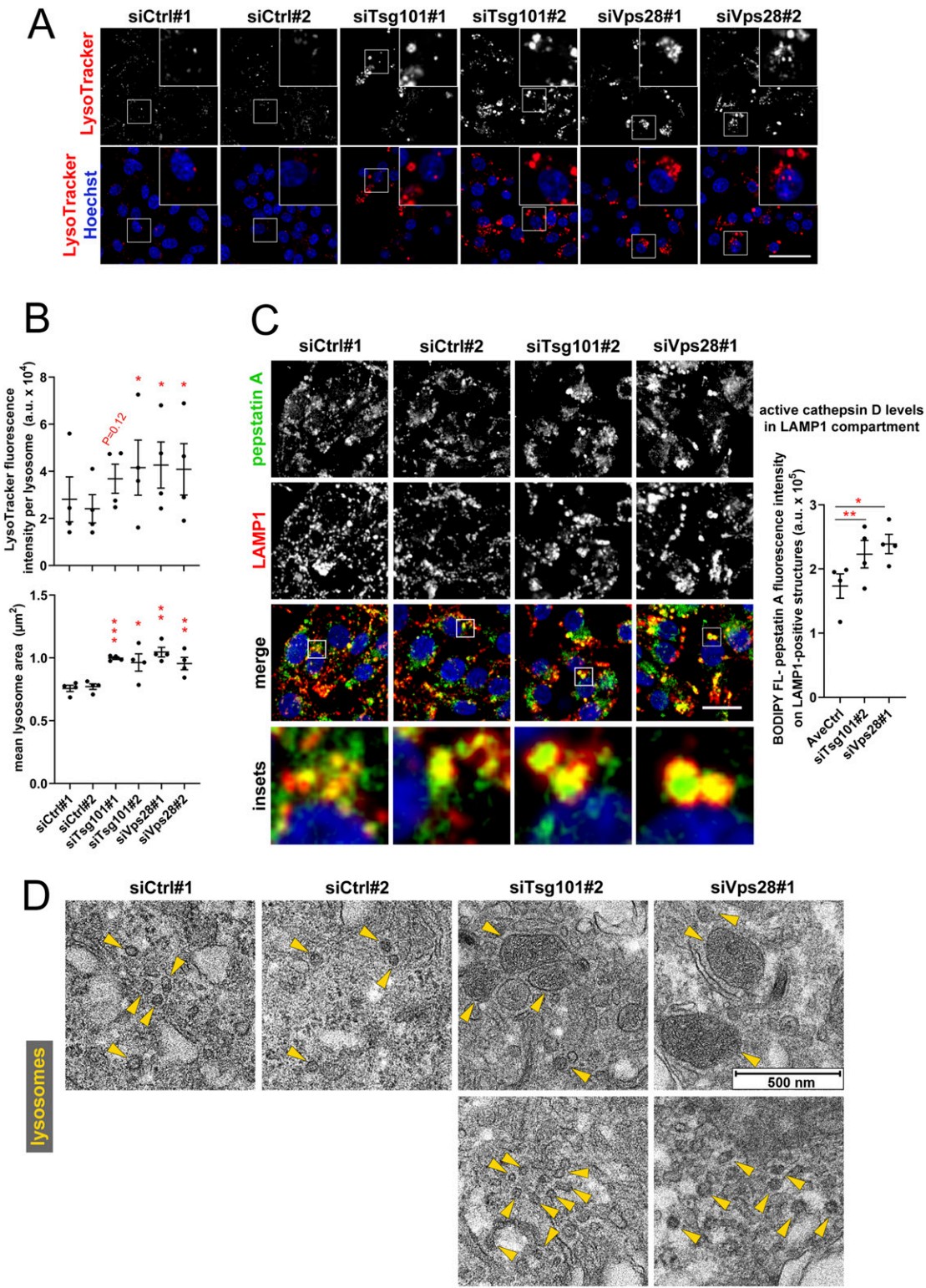

**Figure 1. ESCRT-I dysfunction causes enlargement of lysosomes in RKO cells.**
**(A)** Maximum intensity projection confocal images of live RKO cells, showing the intracellular distribution of lysosomes stained with LysoTracker dye (red) under control conditions (nontargeting siRNAs, Ctrl#1 or #2) and upon depletion of ESCRT-I subunits, Tsg101, or Vps28 (using two single siRNAs for each subunit, #1 or #2). Cell nuclei marked with Hoechst stain (blue). Scale bar, 50 μm. **(A, B)** Dot plots showing average fluorescence intensity of LysoTracker expressed in arbitrary units (a.u.; top panel) and the mean area (bottom panel) of detected lysosomal structures in control or ESCRT-depleted cells, calculated based on live-cell microscopy images (shown in A). Values derived from independent experiments (dots) and their means (n = 4 ± SEM) are presented. Statistical significance tested by comparison to averaged values measured for siCtrl#1 and #2. *P < 0.05, **P < 0.01, ***P < 0.001. **(C)** Maximum intensity projection confocal images of fixed control and ESCRT-I–depleted cells showing

Hence, the absence of ESCRT-I leads to accumulation of enlarged LAMP1-positive acidic structures containing high levels of cathepsin D.

## ESCRT-I limits lysosome size

The resolution of confocal microscopy did not allow determining whether the enlarged LysoTracker-positive structures detected in ESCRT-I–deficient cells represented actual enlarged lysosomes, clusters of closely positioned small lysosomes, or altered morphology of some lysosome-related organelles. Thus, we visualized intracellular organelles using EM. In accordance with the role of ESCRT complexes in sorting of membrane proteins from endosomal limiting membranes into ILVs (Henne et al, 2013), we observed that instead of MVBs that were present in control cells, cells lacking Tsg101 or Vps28 contained numerous enlarged endosomes (Fig S3A). We also noted that cells depleted of ESCRT-I accumulated autophagosomes with nondegraded content (Fig S3A), consistent with the requirement of this complex for proper autophagic degradation (Filimonenko et al, 2007).

When analyzing lysosomes based on their established morphological features, as electron-dense organelles bounded by a monolayer membrane (Aston et al, 2017; de Araujo et al, 2020; Hess & Huber, 2021), we discovered that control RKO cells contained mainly small lysosomes, mostly around 50 nm (Fig 1D), being at the lower end of the typical lysosome size range (50–500 nm) (Bandyopadhyay et al, 2014). We also detected multilamellar bodies (MLBs) (Fig S3B), lysosome-related organelles that contain multiple concentric membrane layers and are involved in lipid storage and secretion (Schmitz & Muller, 1991). In cells lacking ESCRT-I, we detected lysosomes that were markedly bigger than in control cells (Fig 1D), which corroborated our findings using confocal microscopy. However, by EM, we also detected lysosomes similar in size to those in control cells, although concentrated in bigger clusters (Fig 1D). On the contrary, we did not observe any effects of ESCRT-I depletion on the size or morphology of MLBs (Fig S3B).

The EM analysis confirmed that in CRC cells, ESCRT-I limits the size of lysosomes. Based on the abundance of lysosomal markers (shown in Fig 1A–C), we concluded that the enlarged lysosomes from ESCRT-I–deficient cells are not damaged and likely retain their degradative potential.

## ESCRT-I mediates the turnover of lysosomal membrane proteins

We investigated the mechanism through which ESCRT-I controls lysosomal size. As recently shown, degradation of proteins residing in the lysosomal membrane requires their internalization into the lumen of lysosomes together with adjacent membrane parts (Lee et al, 2020). Hence, we hypothesized that the enlargement of lysosomes in the absence of ESCRT-I could be caused by inhibited internalization of parts of lysosomal membranes due to impaired turnover of resident membrane proteins. To address this, we analyzed by confocal microscopy the amount of ubiquitinated proteins on lysosomes using antibodies recognizing mono- and polyubiquitinated protein conjugates. In control RKO cells, we observed a weak ubiquitin staining on the LysoTracker- and LAMP1-positive structures, likely reflecting ubiquitinated cargo targeted for lysosomal degradation (Fig 2A). However, depletion of ESCRT-I subunits led to a strong accumulation of ubiquitin on vesicular structures. This included enlarged lysosomes, marked by LysoTracker staining, in which ubiquitin was particularly enriched on their membranes marked by LAMP1 (Fig 2A). This pointed to an impaired degradation of lysosomal membrane proteins.

To verify the inhibited turnover of proteins from lysosomal membranes upon ESCRT-I depletion, we generated an RKO cell line stably expressing ectopically introduced, GFP-tagged mucolipin1 (GFP-MCOLN1, Fig 2B), a $Ca^{2+}$ channel protein recently shown to be removed from lysosomal membranes by its internalization into the lumen (Lee et al, 2020). As expected (Cheng et al, 2010), we observed by confocal microscopy that in control cells, GFP-MCOLN1 localized predominantly to LAMP1-positive structures, both LysoTracker-negative late endosomes and LysoTracker-positive lysosomes (Fig 2C). Knockdown of ESCRT-I subunits increased the levels of GFP-MCOLN1 protein, observed by Western blotting and microscopy (Fig 2B and C). Importantly, upon ESCRT-I depletion, GFP-MCOLN1 strongly accumulated on LAMP1-positive structures (Fig 2C and D), including enlarged lysosomes (Fig 2C). To address whether the observed accumulation of GFP-MCOLN1 occurred because of impaired lysosomal degradation, we used bafilomycin A1 (BafA1), an inhibitor of the V-ATPase proton pump (Yoshimori et al, 1991). Consistent with its inhibitory effect on lysosomal acidification, BafA1 led to a loss of LysoTracker staining in both control and ESCRT-I–depleted cells (Fig 2C). In control cells, BafA1 caused a strong accumulation of GFP-MCOLN1 protein on LAMP1-positive structures, to the similar levels as observed in ESCRT-depleted cells without BafA1 (Fig 2C and D). However, BafA1 did not significantly increase the amount of GFP-MCOLN1 already enriched on LAMP1-positive structures in cells lacking Tsg101 or Vps28 (Fig 2C and D).

The above results indicated that GFP-MCOLN1 is constantly degraded in lysosomes; however, ESCRT-I depletion inhibits its degradation and thereby causes its accumulation on lysosomal membranes. To verify that the accumulation of GFP-MCOLN1 in cells lacking ESCRT-I occurs because of increased stability of the protein, we inhibited its synthesis using cycloheximide (CHX). As expected, in control cells, treatment with CHX (up to 8 h) reduced GFP-MCOLN1 levels (Fig 2E). However, in cells depleted of Tsg101, the CHX treatment had no effect on the abundance of GFP-MCOLN1 (Fig 2E), confirming the increased protein stability.

Thus, our data show that ESCRT-I mediates the lysosomal degradation of late endosomal and lysosomal membrane proteins,

intracellular distribution of cathepsin D, stained with pepstatin A conjugated to BODIPY FL (green), as compared to late endosomes/lysosomes, detected using anti-LAMP-1 antibody (red). Cell nuclei marked with DAPI stain (blue). Scale bar, 20 $\mu$m. The dot plot on the right shows average BODIPY FL fluorescence intensity of LAMP-1–positive structures calculated based on confocal images. Values derived from independent experiments and their means (n = 4 ± SEM) are presented. Statistical significance tested by comparison to averaged value measured for siCtrl-treated cells (AveCtrl). *P < 0.05, **P < 0.01. **(D)** Representative EM images of control and ESCRT-I–depleted cells showing the morphology and size of lysosomes (indicated by yellow arrowheads). Scale bar, 500 nm.

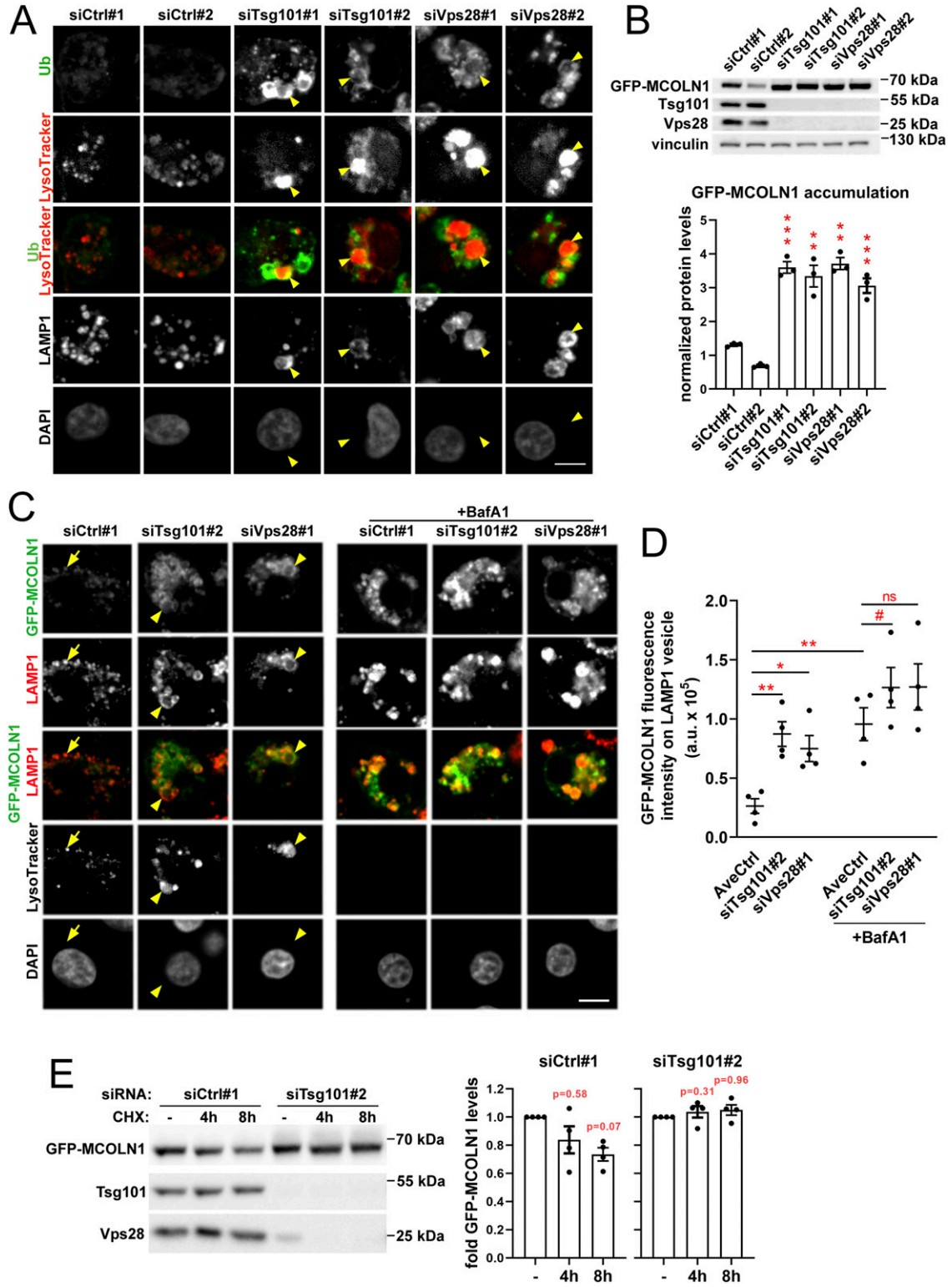

**Figure 2. ESCRT-I mediates the degradation of lysosomal membrane proteins in RKO cells.**
**(A)** Maximum intensity projection confocal images of fixed RKO cells showing intracellular distribution of ubiquitin (green), LysoTracker dye (red), and LAMP1 (gray) in ESCRT-I–depleted (siTsg101#1 or #2, siVps28#1 or #2) or control (Ctrl#1, #2, nontargeting siRNAs) cells. Cell nuclei marked with DAPI stain (gray). Enlarged lysosomes enriched in ubiquitinated proteins at lysosomal outer membranes, marked with LAMP1, are indicated by arrowheads. Scale bar, 10 μm. **(B)** Representative Western blots (upper panel) showing the levels of ectopically expressed GFP-MCOLN1 (detected by anti-GFP antibodies) and ESCRT-I components in control or ESCRT-I–depleted RKO cells. The graph (lower panel) shows GFP-MCOLN1 levels expressed as fold change with respect to averaged values measured for siCtrl#1 and #2 by densitometry analysis of Western blotting bands. Vinculin was used as a gel-loading control. Values derived from independent experiments and their means (n = 4 ± SEM) are presented.

including MCOLN1. This discovery corroborates our hypothesis that the enlargement of lysosomes in CRC cells lacking ESCRT-I could stem from inhibited turnover of lysosomal membrane proteins.

## ESCRT-I deficiency activates transcription of genes involved in lysosomal biogenesis and cholesterol biosynthesis

The involvement in turnover of lysosomal membrane proteins is yet another function of ESCRT-I, alongside endosomal sorting and autophagosome maturation, in delivering cargo to lysosomal degradation. Hence, we reasoned that the absence of ESCRT-I could impair the acquisition of metabolites from lysosomes, which would be reflected by the activation of specific transcriptional responses. To address whether such transcriptional responses occur because of ESCRT-I depletion in RKO cells, we performed RNA-sequencing. Gene ontology analysis of commonly up-regulated genes after Tsg101 or Vps28 depletion identified an elevated inflammatory response, consistent with our previous reports (Maminska et al, 2016; Banach-Orlowska et al, 2018). Importantly, we also observed an enhanced expression of genes annotated to autophagy and cholesterol metabolism (Fig 3A), two processes highly dependent on lysosomes (Ikonen, 2008; Sardiello et al, 2009; Lieberman et al, 2012; Medina et al, 2015).

Among genes annotated to autophagy, whose expression was induced upon ESCRT-I depletion, we identified a group of genes involved in lysosomal biogenesis and autolysosomal degradation that are established targets of MiT-TFE transcription factors (Fig 3B), known to be activated because of starvation or lysosomal stress (Settembre et al, 2012; Medina et al, 2015; Perera & Zoncu, 2016). Among genes annotated to cholesterol metabolism were mainly those encoding enzymes of cholesterol biosynthesis (Fig 3C), whose expression is typically induced upon impaired delivery to the ER of cholesterol provided by the endolysosomal trafficking (Luo et al, 2020; Xue et al, 2020).

Hence, in addition to the previously reported induction of inflammatory signaling (Maminska et al, 2016; Kolmus et al, 2021), ESCRT-I depletion activates transcriptional responses suggestive of lysosomal stress or impaired delivery of nutrients from autolysosomal and endolysosomal compartments.

## Depletion of ESCRT-I leads to a prolonged activation of TFEB and TFE3 transcription factors

To validate the transcriptomic data suggesting activation of MiT-TFE transcription factors, we tested whether ESCRT-I deficiency affected the localization of TFEB and TFE3 proteins. Consistent with their reported activation in various cancer types, including colon cancer

(Astanina et al, 2021), we detected basal amounts of TFEB and TFE3 in nuclear fractions of control RKO cells by Western blotting analysis (Fig 4A). Nevertheless, we observed that depletion of Tsg101 or Vps28 markedly increased the nuclear abundance of both factors (Fig 4A). We also found increased TFEB/TFE3 nuclear translocation upon ESCRT-I deficiency in other cell lines, DLD-1 and HEK293 (Fig S4A and B), pointing to a common intracellular response. Next, we verified whether the induced expression of genes involved in the regulation of lysosomal function in ESCRT-I–depleted RKO cells occurred due to the MiT-TFE activation. To this end, we depleted TFEB and TFE3 factors using siRNA. We observed that their simultaneous depletion in cells lacking Tsg101 or Vps28 prevented the elevated expression of two MiT-TFE target genes that encode lysosomal proteins, NPC1 and MCOLN1 (Fig 4B). The activation of TFEB/TFE3-dependent expression of lysosome biogenesis genes likely accounts for the accumulation of new small lysosomes in ESCRT-I–deficient cells shown in Fig 1D.

Although our results revealing elevated MiT-TFE signaling upon ESCRT-I depletion (shown in Figs 3B and 4A and B) were obtained 72 h post transfection (hpt) of cells with siRNA, we suspected that activation of these transcription factors could occur earlier. To this end, by quantitative analysis of confocal microscopy images, we measured the percentage of RKO cells with TFEB or TFE3 present in their nuclei. We detected nuclear TFEB or TFE3 in around 10% of control cells, a fraction that was constant at different time-points post transfection (Fig 4C), likely accounting for basal levels of TFEB and TFE3 in nuclear fractions shown in Fig 4A. However, upon Tsg101 depletion, the percentage of RKO cells with nuclear accumulation of these transcription factors increased significantly already at 48 hpt, in the case of TFEB to the similar level (around 30%) as observed at 72 hpt (Fig 4C).

Taken together, we verified that cells lacking ESCRT-I activate TFEB/TFE3 signaling, which reinforced our hypothesis that ESCRT-I deficiency leads to lysosomal dysfunction or impaired delivery of nutrients through lysosome-dependent processes.

## ESCRT-I is required for proper endolysosomal transport of cholesterol

The induced transcription of genes involved in cholesterol biogenesis (shown in Fig 3C) is indicative of cholesterol deficiency (Trinh et al, 2020). Thus, we reasoned that cholesterol could be one of nutrients that are inefficiently supplied via endolysosomal trafficking in cells lacking ESCRT-I. In accordance with this, we observed that depletion of Tsg101 or Vps28 in RKO cells led to the accumulation of cholesterol (stained by filipin) in enlarged LAMP1-positive structures (Fig 5A). This accumulation was suppressed when cells were cultured in a delipidated medium (Fig 5A) that has

---

Statistical significance tested by comparison to siCtrl#1. **$P < 0.01$, ***$P < 0.001$. **(C)** Representative single confocal plane images of fixed cells showing the effect of ESCRT-I depletion and/or 18 h bafilomycin A1 (BafA1) treatment on the intracellular distribution of ectopically expressed GFP-MCOLN1 (green), with respect to LAMP1 (red) and LysoTracker dye (gray). In control cells, GFP-MCOLN1 accumulation on LAMP1-positive vesicles indicated by arrows. GFP-MCOLN1 accumulation on enlarged LAMP1-positive lysosomal structures in ESCRT-I–depleted cells indicated by arrowheads. Cell nuclei marked with DAPI stain (gray). Scale bar, 10 $\mu m$. **(C, D)** Dot plot showing the fluorescence intensity of GFP-MCOLN1 colocalizing with LAMP1-positive vesicles expressed in arbitrary units (a.u.) in confocal microscopy images of control or ESCRT-I–depleted cells and/or upon 18 h bafilomycin A1 (BafA1) treatment (shown in C). Values derived from independent experiments (dots) and their means (n = 4 ± SEM) are presented. Statistical significance tested by comparison to averaged values measured for siCtrl#1 and #2 (AveCtrl). ns—nonsignificant, #$P < 0.1$, *$P < 0.05$, **$P < 0.01$. **(E)** Representative Western blots showing the abundance of GFP-MCOLN1, Tsg101, and Vps28 proteins in control and Tsg101-depleted cells treated with cycloheximide (CHX, 100 $\mu g$/ml) for the indicated time periods. Graphs on the right show fold change of GFP-MCOLN1 levels measured by densitometry analysis of Western blotting bands, including those shown on the left. Values derived from independent experiments and their means (n = 4 ± SEM) are presented. Statistical significance tested by comparison to siCtrl#1- or siTsg101#2-treated samples.

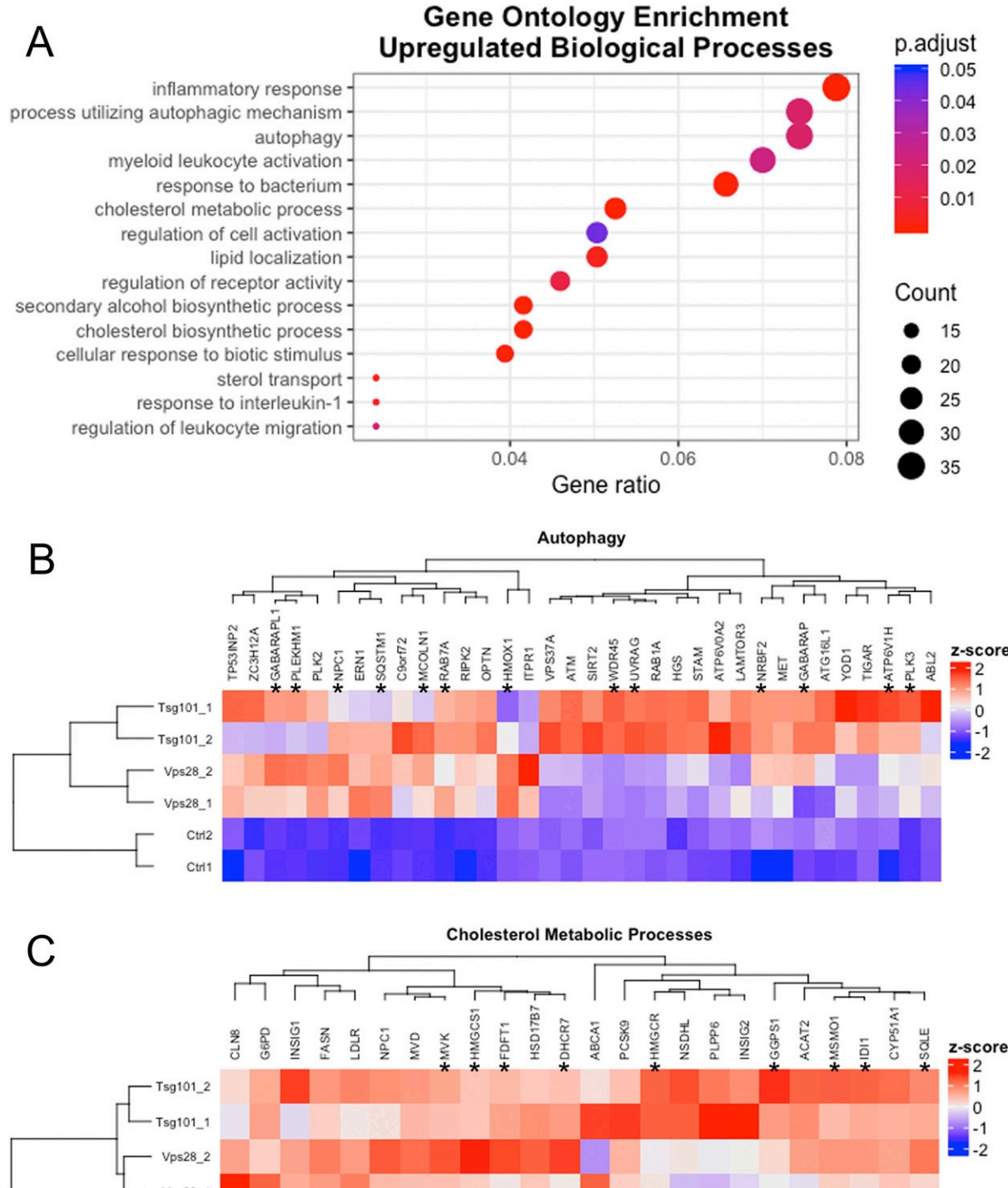

**Figure 3. Depletion of ESCRT-I induces the expression of genes annotated to autophagy and cholesterol biosynthesis.**
**(A)** Gene ontology (GO) analysis of top biological processes identified by annotation of genes with up-regulated expression (≥1.5-fold; adjusted *P*-value < 0.05), detected by RNA-Seq in RKO cells depleted of Tsg101 or Vps28 (siTsg101#1 or #2, siVps28#1 or #2), as compared to control cells (treated with nontargeting siRNAs, Ctrl#1 or #2). RNA-Seq data analysis was performed based on three independent experiments. **(B)** Heatmap visualizing expression of genes annotated to "autophagy" (GO:0006914) process, whose mRNA levels were detected by RNA-Seq as up-regulated after Tsg101 or Vps28 depletion in RKO cells. Established MiT-TFE target genes are indicated by asterisks.
**(C)** Heatmap visualizing expression of genes annotated to "cholesterol metabolic processes" (GO:0008203) process, whose mRNA levels were detected by RNA-Seq as up-regulated after Tsg101 or Vps28 depletion in RKO cells. Established cholesterol biosynthesis genes are indicated by asterisks.

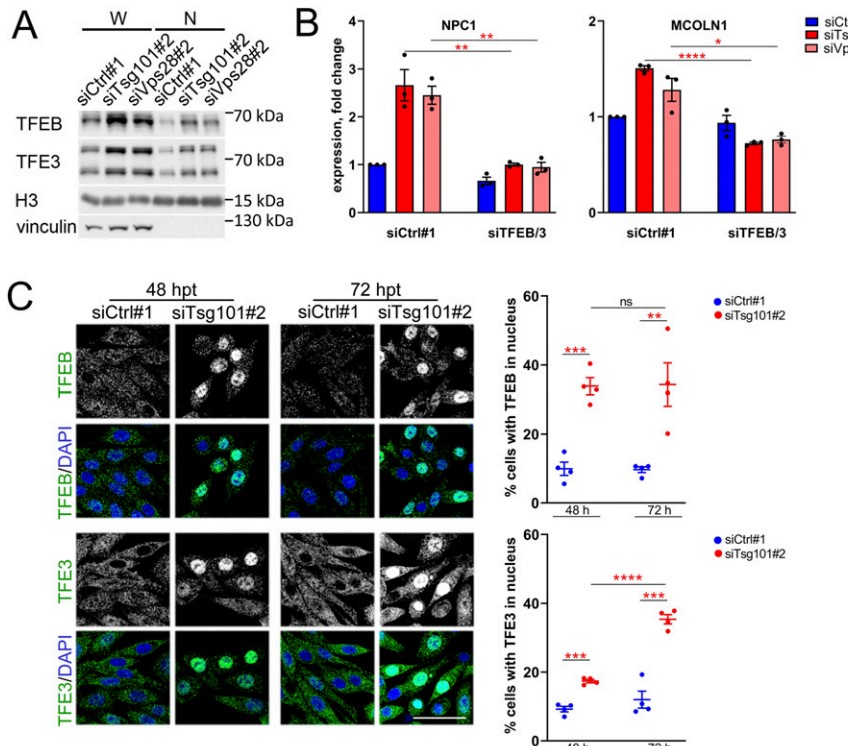

**Figure 4. Depletion of ESCRT-I induces prolonged activation of TFEB/TFE3 signaling.**
**(A)** Western blots showing levels of TFEB and TFE3 proteins in whole-cell lysates (W) and nuclear fractions (N) of RKO cells depleted of Tsg101 or Vps28 (siTsg101#2, siVps28#2), as compared to control cells (treated with nontargeting siRNA, Ctrl#1). To examine the fraction purity, the levels of vinculin (cytosolic marker) were probed. The H3 histone protein was used as a loading control for nuclear fractions. **(B)** qPCR results showing the expression of MiT-TFE target genes upon ESCRT-I and/or MiT-TFE depletion (using single siRNAs for Tsg101, Vps28, TFEB, or TFE3) presented as fold changes with respect to control cells. Values derived from independent experiments and their means (n = 4 ± SEM) are presented. Statistical significance tested by comparison to siTsg101 or siVps28 conditions. *P < 0.05, **P < 0.01, ****P < 0.0001. **(C)** Maximum intensity projection confocal images of fixed RKO cells at 48 or 72 h post transfection (hpt) showing intracellular distribution of TFEB or TFE3 (green) in ESCRT-I–depleted or control RKO cells. Cell nuclei marked with DAPI (blue). Dot plots on the right show percentage of cells with nuclear TFEB or TFE3 localization. Values derived from independent experiments (dots) and their means (n = 4 ± SEM) are presented. Statistical significance tested by comparison to siCtrl#1 and/or siTsg101#2 conditions. ns, nonsignificant (P ≥ 0.05), **P < 0.01, ***P < 0.001, ****P < 0.0001.

reduced levels of lipids, including cholesterol which under such conditions is not taken up via endolysosomal trafficking (Brovkovych et al, 2019). These data argue that cholesterol accumulated in enlarged LAMP1-positive structures upon ESCRT-I depletion is of extracellular origin.

To verify whether the up-regulated expression of cholesterol biosynthesis genes in the absence of ESCRT-I (shown in Fig 3C) is due to an impaired delivery of cholesterol from late endosomes or lysosomes to the ER, we supplied cells with cholesterol in a soluble form that reaches the ER independently of the endolysosomal trafficking (Trinh et al, 2020). As analyzed by quantitative RT-PCR, soluble cholesterol supplementation prevented the up-regulation of cholesterol biosynthesis genes in RKO cells lacking Tsg101 or Vps28 (Fig S5). Hence, we confirmed an impaired delivery of cholesterol from endolysosomal compartments to the ER in the absence of ESCRT-I.

### Endolysosomal accumulation of cholesterol does not contribute to the activation of TFEB/TFE3 signaling in ESCRT-I–deficient cells

The MiT-TFE–dependent transcriptional responses observed in cells lacking ESCRT-I did not allow discriminating whether these cells activate stress signaling, due to dysfunction of the enlarged lysosomes or starvation-like signaling, due to impaired nutrient delivery through lysosome-dependent processes. The MiT-TFE signaling is regulated by various intracellular cues that may reflect lysosomal stress or nutrient deficiency. These cues include (i) accumulation of cholesterol in lysosomes causing cholesterol-induced lysosomal stress (Willett et al, 2017), (ii) calcium-dependent signaling

that has been linked to both starvation response and integrated stress response (Medina et al, 2015; Martina et al, 2016), and (iii) inactivation of Rag GTPase–dependent mTORC1 signaling that responds to nutrient delivery from lysosomes (Martina & Puertollano, 2013). Therefore, to understand the consequences of ESCRT-I depletion for lysosomal function, we investigated the molecular mechanisms that underlie the observed nuclear translocation of TFEB and TFE3.

First, we tested whether preventing the retention of cholesterol within the endolysosomal pathway by culturing cells in a delipidated medium affected the induction of TFEB/TFE3 signaling at its early stage (48 hpt). Intriguingly, we observed that deprivation of exogenous lipids strongly reduced the percentage of control cells with nuclear TFEB or TFE3 (Fig 5B), indicating that basal activation of this pathway depends on lipid delivery. However, culture of cells in the absence of exogenous lipids did not prevent the increase in the number of cells with nuclear TFEB or TFE3 upon Tsg101 depletion.

Thus, we found that the accumulation of cholesterol in the endolysosomal system upon ESCRT-I deficiency is not a causative factor for the induction of TFEB/TFE3 signaling, arguing against the possibility of cholesterol-induced lysosomal stress occurring under these conditions.

### Calcium signaling is required for activation of TFEB and TFE3 transcription factors but does not mediate TFEB dephosphorylation upon ESCRT-I depletion

We next explored the second possible mechanism of activated TFEB/TFE3 signaling upon ESCRT-I depletion that could be related

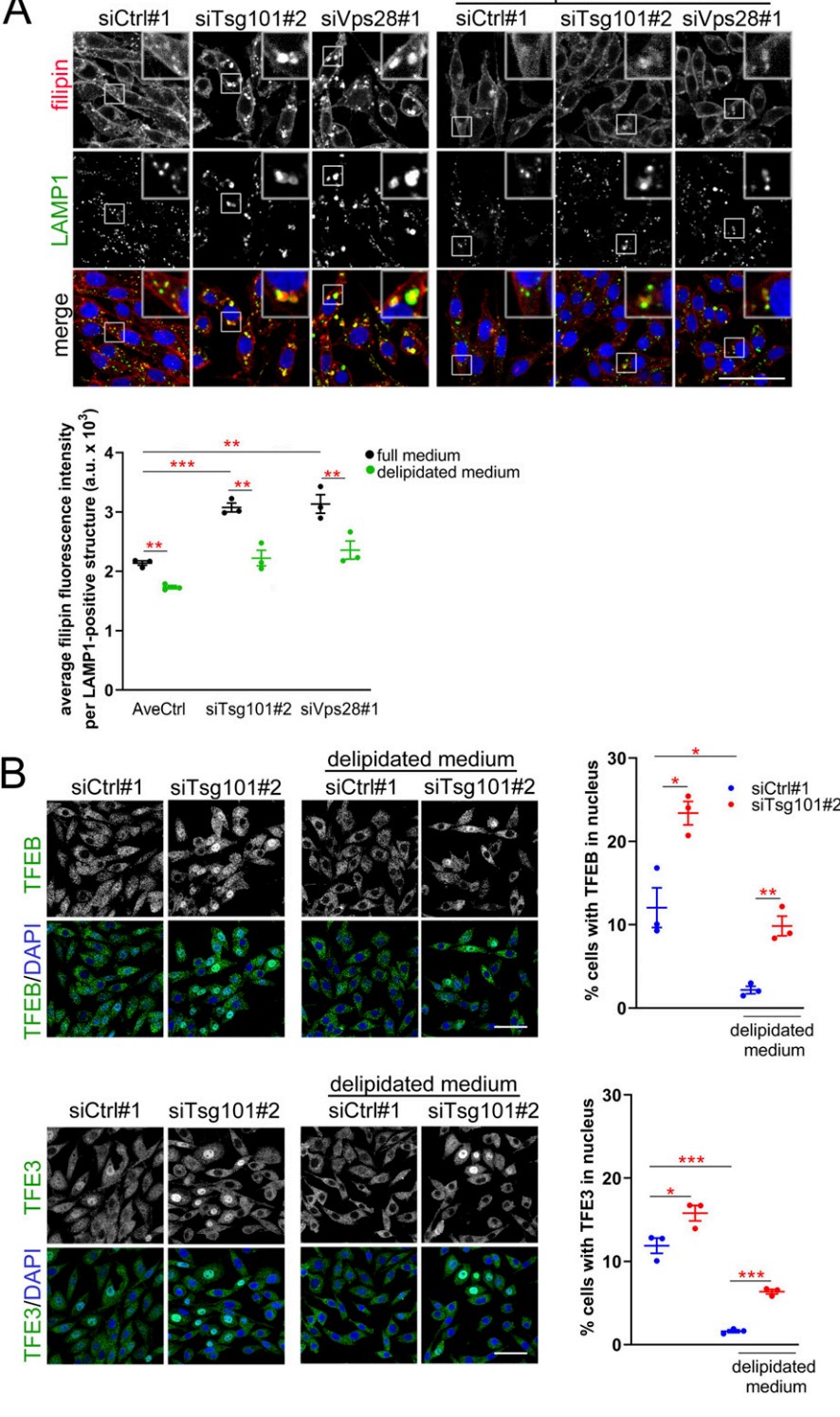

**Figure 5. ESCRT-I deficiency leads to endolysosomal accumulation of cholesterol that does not contribute to activation of TFEB/TFE3 signaling. (A)** Maximum intensity projection confocal images of fixed RKO cells at 48 h post transfection (48 hpt) showing the effect of culture in delipidated medium (for 40 h) on the intracellular distribution of free cholesterol marked with filipin dye (red) with respect to LAMP1 protein (green) in control (treated with nontargeting siRNA, Ctrl#1) or ESCRT-I–depleted (siTsg101#2 or siVps28#1) cells. Cell nuclei marked with DRAQ7 dye (blue). Scale bar, 50 $\mu$m. The dot plot on the bottom shows the average filipin fluorescence intensity per LAMP1-positive structure (expressed in arbitrary units, a.u.), as compared to control cells. Values derived from independent experiments (dots) and their means (n = 3 ± SEM) are presented. Statistical significance tested by comparison to averaged values measured for control cells (AveCtrl) and/or siTsg101#2 conditions. **$P < 0.01$, ***$P < 0.001$. **(B)** Maximum intensity projection confocal images of fixed RKO cells at 48 hpt showing the effect of culture in delipidated medium on the intracellular distribution of TFEB or TFE3 (green) in control or ESCRT-I–depleted cells. Cell nuclei marked with DAPI (blue). Dot plots on the right show the percentage of cells with nuclear TFEB or TFE3 localization. Values derived from independent experiments (dots) and their means (n = 3 ± SEM) are presented. Statistical significance tested by comparison to siCtrl#1 conditions. *$P < 0.05$, **$P < 0.01$, ***$P < 0.001$.

to MCOLN1-Ca$^{2+}$-calcineurin signaling, encouraged by our observation of impaired turnover of MCOLN1, a Ca$^{2+}$ channel under these conditions (Fig 2B–E). To this end, we applied BAPTA-AM, a chelator of intracellular Ca$^{2+}$ (Medina et al, 2015); ML-SI1, an inhibitor of MCOLN1 channel activity (Sun et al, 2018); or cyclosporin A (CsA), an inhibitor of calcineurin (Liu et al, 1991) at an early stage of the pathway activation (48 hpt). Surprisingly, treatment with these compounds did not decrease the nuclear levels of TFEB and TFE3 in control cells, indicating that basal activation of MiT-TFE factors is not mediated by Ca$^{2+}$ signaling (Figs 6A and S6). Conversely, all

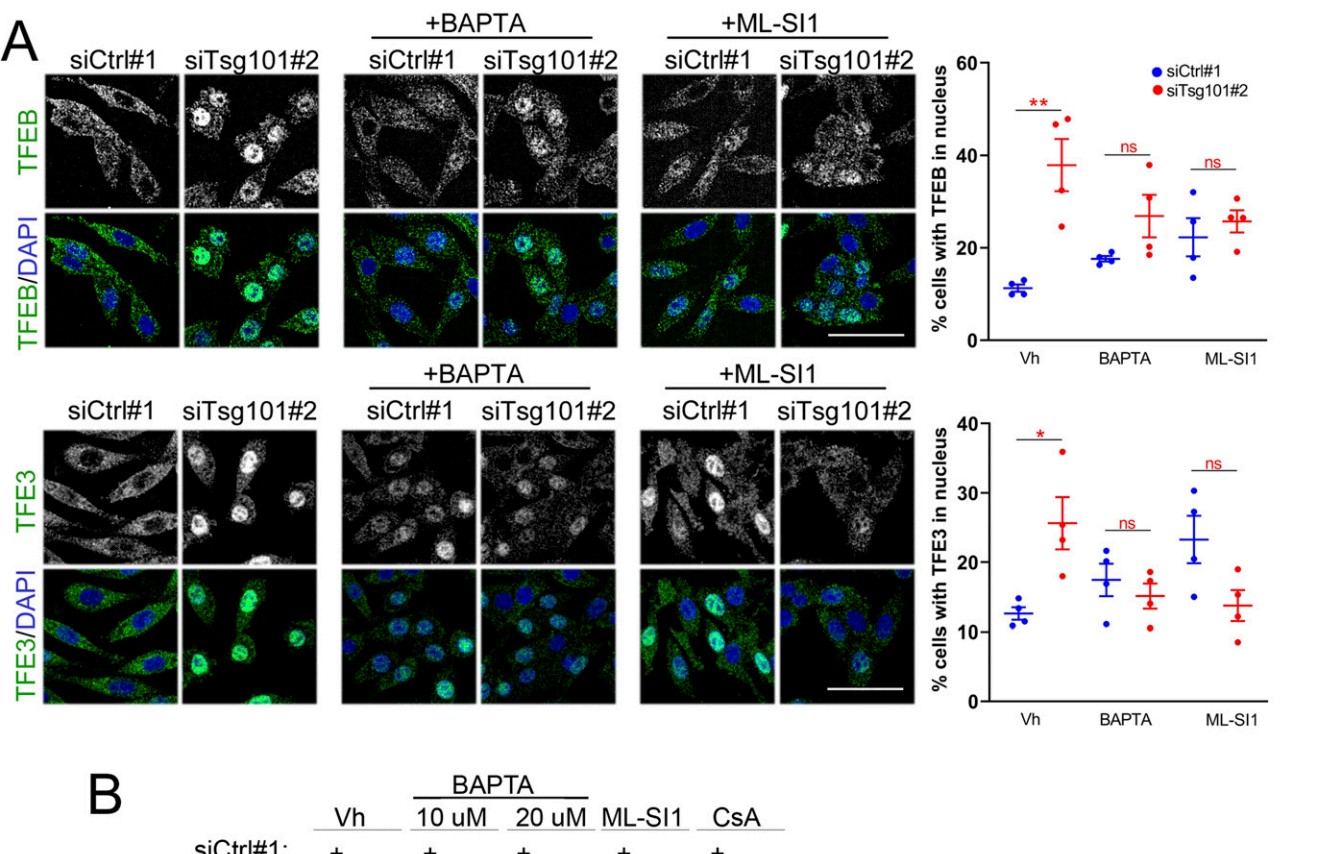

**Figure 6. Ca²⁺-dependent signaling is required for TFEB/TFE3 nuclear localization but not for TFEB dephosphorylation in cells lacking Tsg101.**
**(A)** Maximum intensity projection confocal images of fixed RKO cells at 48 h post transfection (48 hpt) showing intracellular distribution of TFEB or TFE3 (green) in ESCRT-I–depleted (siTsg101#2) or control (Ctrl#1, nontargeting siRNA) cells upon 2 h treatment with vehicle (Vh, DMSO), BAPTA-AM (10 $\mu$M), or ML-SI1 (25 $\mu$M). Cell nuclei marked with DAPI stain (blue). Scale bar, 50 $\mu$m. Dot plots on the right show the percentage of cells with nuclear TFEB or TFE3 localization. Values derived from independent experiments (dots) and their means (n = 4 ± SEM) are presented. Statistical significance tested by comparison to siCtrl#1 conditions. ns, nonsignificant, *P < 0.05, **P < 0.01. **(B)** Representative Western blots showing levels of phosphorylated (at Ser122 or Ser211) or total TFEB and of Tsg101 and vinculin (loading control) in lysates obtained 48 hpt from control (siCtrl#1) or Tsg101-depleted RKO cells treated for 3 h with the indicated reagents: vehicle (Vh, DMSO), 10 or 20 $\mu$M BAPTA, 25 $\mu$M ML-SI1 or 25 $\mu$M CsA.

these compounds had a tendency to increase the TFEB/TFE3 nuclear levels in control cells. However, Ca²⁺ chelation or inhibition of MCOLN1 or calcineurin prevented the nuclear accumulation of TFEB and TFE3 proteins because of Tsg101 depletion (Figs 6A and S6), indicating that activation of MiT-TFE factors upon ESCRT-I deficiency requires Ca²⁺-dependent signaling.

To investigate whether the activation of MiT-TFE factors upon ESCRT-I deficiency involves their Ca²⁺-induced dephosphorylation, we analyzed by Western blotting in RKO cells two phosphorylation

sites of TFEB, at serine 211 and 122, that cooperate to inhibit its nuclear translocation (Vega-Rubin-de-Celis et al, 2017). Although in control cells we could repeatedly detect a clear signal for the S122 phosphorylation, the S211 phosphorylation was barely detected (Fig 6B). Consistent with its effect on induction of TFEB nuclear translocation, Tsg101 depletion reduced the levels of both phosphorylations (Fig 6B). Surprisingly, inhibition of MCOLN1-Ca²⁺-calcineurin signaling by various compounds, which we expected to raise phosphorylation levels of at least the S211 site (Medina et al,

2015; Zhang et al, 2016), inhibited both phosphorylations in control cells and did not restore the phosphorylation levels reduced by Tsg101 depletion (Fig 6B).

Collectively, we observed that calcium signaling is required for inducing TFEB/TFE3 nuclear translocation upon ESCRT-I depletion. Yet, the underlying mechanism of this regulation does not involve the canonical $Ca^{2+}$-dependent dephosphorylation by calcineurin.

### ESCRT-I deficiency reduces TFEB phosphorylation at S122 without having a broad effect on mTORC1 signaling

After excluding the involvement of endolysosomal cholesterol accumulation or $Ca^{2+}$-dependent dephosphorylation, we investigated whether the induction of TFEB/TFE3 in ESCRT-I depleted cells occurs via regulation of mTORC1 kinase signaling. Having observed reduced phosphorylation of S122 that is a direct mTORC1 target (Vega-Rubin-de-Celis et al, 2017), we investigated whether it occurs because of inhibition of general mTORC1 signaling, which would point to a broad starvation response in ESCRT-I–deficient cells (Ng et al, 2011). To this end, we compared the regulation of TFEB S122 phosphorylation to that of other described mTORC1 kinase targets (Ulk1, S6K, or 4E-BP1), in control or ESCRT-I–depleted cells, under normal growth conditions (EMEM full medium) or upon nutrient deprivation (EBSS medium). In control cells, the phosphorylation signals of all tested targets were easily detected upon EMEM, and their levels were clearly reduced upon EBSS (Fig 7A and B), verifying that all of these phosphorylations are under constant activation in RKO cells. Importantly, the reduction of S122 phosphorylation due to depletion of Tsg101 or Vps28 was as strong as observed for control cells upon EBSS (Fig 7A and B). However, ESCRT-I deficiency did not inhibit phosphorylations of other tested mTORC1 targets (Fig 7A and B). Reassuringly, we also observed reduced phosphorylation of TFEB at S122 but not of Ulk1, S6K, or 4E-BP1 in cells lacking ESCRT-I because of CRISPR/Cas9–mediated Tsg101 depletion (Fig S7A; depletion efficiencies shown in Fig S1C).

Consistent with no effect on the canonical mTORC1 targets, we found that ESCRT-I depletion did not alter the association of the mTOR protein with LAMP1-positive structures (Fig S7B) that is required for general mTORC1 signaling (Sancak et al, 2010), either upon normal growth conditions or when this association was reduced due to nutrient deprivation.

Hence, we discovered a specific response to ESCRT-I deficiency that involves reduced S122 phosphorylation of TFEB, independent of canonical mTORC1 signaling. This pointed to the inhibition of a recently identified mTORC1 pathway that specifically regulates MiT-TFE factors (Napolitano et al, 2020; Alesi et al, 2021).

### ESCRT-I restricts RagC-dependent mTOR-TFEB/TFE3 signaling

The mTORC1-dependent inhibition of TFEB nuclear translocation is mediated by the Rag GTPase complex activated in response to lysosomal amino acids (Martina & Puertollano, 2013; Settembre et al, 2012). Hence, we hypothesized that cells lacking ESCRT-I activate TFEB/TFE3 signaling via inhibition of Rag GTPase activity, which would indicate shortage of lysosome-derived nutrients. In the presence of nutrients, Rag GTPases promote the lysosomal recruitment of MiT-TFE factors, which can be observed upon

pharmacological inhibition of mTOR kinase (Martina & Puertollano, 2013). To test the status of the Rag GTPase–dependent mTORC1 pathway in ESCRT-I–deficient cells, we investigated the colocalization of TFEB and TFE3 proteins with LAMP1-positive structures (TFE-LAMP1 colocalization) in vehicle-treated cells or cells with inhibition of mTOR kinase using the INK128 compound. As expected, INK128 treatment induced TFEB/TFE3 nuclear translocation that was associated with a marked increase in TFE-LAMP1 colocalization as compared to control, vehicle-treated RKO cells (Figs 7C–E and S7C–E). Conversely, activation of TFEB/TFE3 signaling because of Tsg101 depletion was not associated with any increase in TFE-LAMP1 colocalization (Figs 7C–E and S7C–E). Moreover, lack of Tsg101 partially prevented the INK128-induced lysosomal recruitment of TFEB or TFE3 (Figs 7C–E and S7C–E). As similar effect was shown for amino acid starvation (Martina & Puertollano, 2013), these results supported the hypothesis that ESCRT-I depletion causes a shortage of lysosome-derived nutrients.

We noted that in cells lacking Tsg101, INK128 treatment only modestly increased TFEB nuclear levels (Fig 7C and D) and had no effect on TFE3 nuclear abundance (S7C-D). This reinforced our hypothesis that ESCRT-I deficiency could activate TFEB/TFE3 signaling by a similar mechanism as mTORC1 inactivation, that is, by inhibition of the Rag GTPase–dependent substrate-specific mTORC1 pathway. To definitively verify this hypothesis, we analyzed the effect of activating the Rag GTPase–dependent pathway on the induction of TFEB/TFE3 signaling in ESCRT-deficient cells. To this end, we overexpressed wild-type (WT) RagC (as a control) or constitutively active RagC mutant (S75L) (Fig 7F). The overexpression of WT RagC had no effect on the basal nuclear abundance of TFEB or TFE3 (again around 10% cells with nuclear staining were detected) and did not prevent their nuclear localization induced upon Tsg101 depletion. However, overexpression of the active RagC mutant inhibited basal TFEB/TFE3 signaling and prevented its induction in cells lacking Tsg101. Importantly, the inhibition of TFEB/TFE3 nuclear translocation by the active RagC mutant, in control and Tsg101-depleted cells, was associated with a strong induction of TFEB S122 phosphorylation (Fig 7G), although levels of Ulk1 or S6K phosphorylations remained unaffected. This underscored the importance of the substrate-specific mTORC1 pathway in regulating both basal TFEB/TFE3 signaling and its induction due to ESCRT-I deficiency.

Based on these findings, we conclude that ESCRT-I deficiency induces TFEB/TFE3 signaling by inactivation of the Rag GTPase–dependent substrate-specific mTORC1 pathway. We propose that this mechanism reflects the shortage of lysosome-derived nutrients, that is, lysosomal nutrient starvation that occurs because of inefficient delivery of cargo for degradation in lysosomes.

## Discussion

Lysosomes have recently emerged as important players in cancer development or progression, and targeting their function has been proposed as a promising strategy in cancer treatment (Tang et al, 2020; Machado et al, 2021). Recently, we showed that components of the ESCRT machinery could serve as potential therapeutic targets

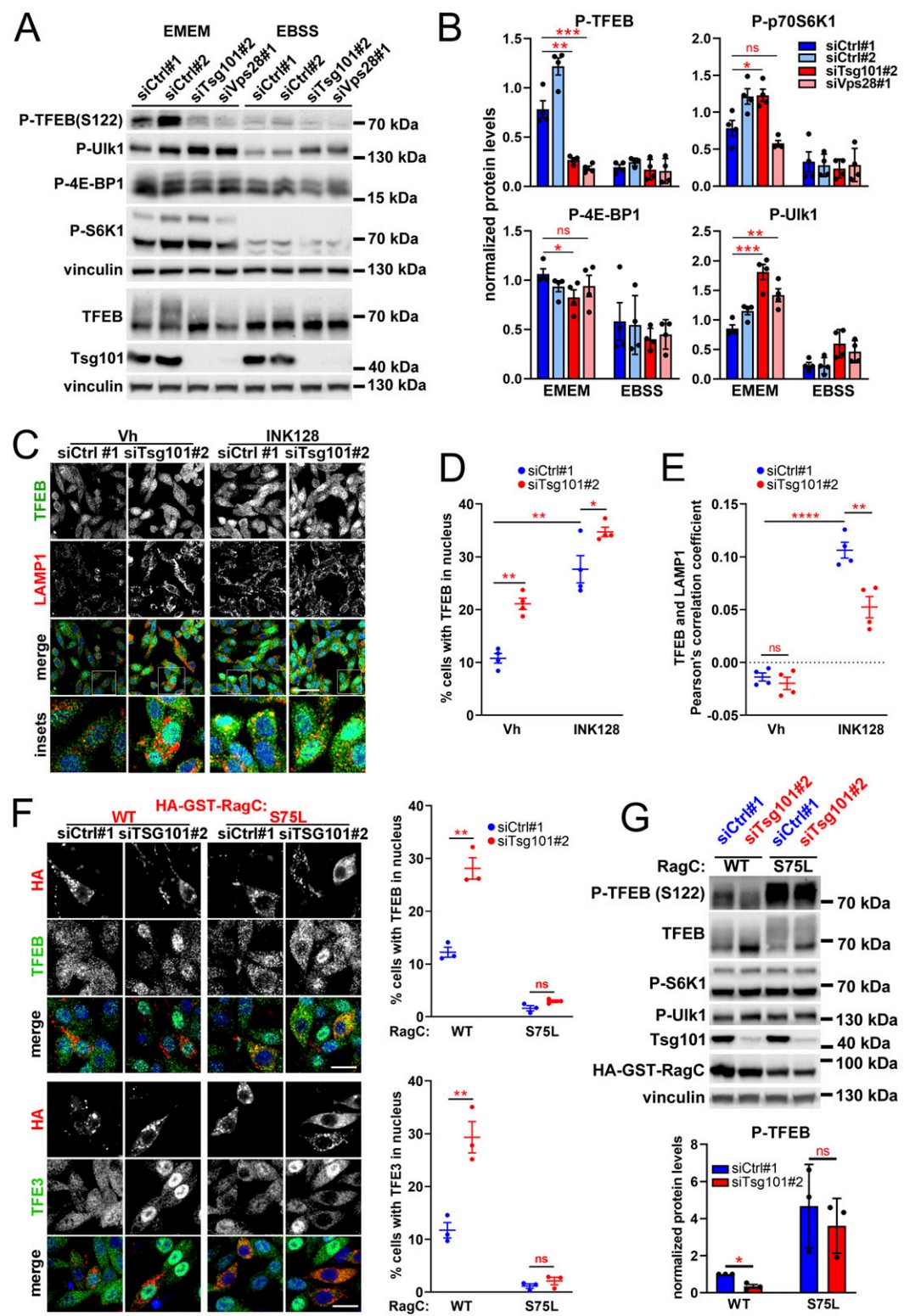

**Figure 7. TFEB/TFE3 signaling in ESCRT-I–deficient cells is activated due to the inhibition of the Rag GTPase–dependent mTORC1 pathway.**
**(A)** Western blots showing levels of phosphorylation of TFEB at Ser122 and the indicated canonical mTORC1 targets in cells lacking ESCRT-I (transfected with siTsg101#2 or siVps28#1) as compared with control cells (siCtrl#1 or #2, nontargeting siRNAs) at 48 h post transfection (48 hpt) upon culture in regular medium (EMEM) or nutrient-deficient medium (EBSS) for 2 h. Vinculin was used as a loading control. **(A, B)** Graphs showing fold levels of indicated phosphorylations measured by densitometry analysis of Western blotting bands, including those shown in (A). Values derived from independent experiments (dots) and their means (n = 4 ± SEM) are presented. Statistical significance tested by comparison to siCtrl#1. ns, nonsignificant, *P < 0.05, **P < 0.01, ***P < 0.001. **(C)** Single-plane confocal images of fixed RKO cells at 48 hpt

against colorectal cancer (CRC) (Szymanska et al, 2020; Kolmus et al, 2021), but whether modulation of ESCRT activity would affect lysosomal function in CRC cells has not been tested. Here, by a comprehensive analysis of the morphology and function of lysosomes in CRC cells, we discover that mammalian ESCRT-I proteins, Tsg101 or Vps28, control lysosomal size and homeostasis, regulating lysosomal membrane protein turnover and endolysosomal transport of cholesterol. Because of the involvement of ESCRT-I in lysosomal homeostasis, the absence of this complex activates transcriptional responses resembling starvation-induced signaling.

Although ESCRT-dependent degradation of vacuolar transmembrane proteins has been shown to occur in yeast (Zhu et al, 2017; Morshed et al, 2020; Yang et al, 2020; Yang et al, 2021), an analogous mechanism has not been fully characterized in mammalian cells. So far, mammalian ESCRT proteins were shown to associate with lysosomes to repair their membranes upon damage (Radulovic et al, 2018; Skowyra et al, 2018; Eriksson et al, 2020). Our study extends these findings by showing that ESCRT-I has a broader function as it maintains lysosomal homeostasis. This is consistent with a very recent discovery of Zhang et al who uncovered that ESCRT components, particularly ESCRT-III and Vps4, are involved in degradation of several lysosomal membrane proteins in HEK293 and HeLa cells (Zhang et al, 2021).

Previous analyses of LAMP1-positive compartment suggested that depletion of ESCRT subunits might affect lysosomal morphology (Doyotte et al, 2005; Du et al, 2012; Du et al, 2013; Szymanska et al, 2020). However, as LAMP1 is a marker of both late endosomes and lysosomes (Saftig & Klumperman, 2009), these observations did not allow conclusions about the status of lysosomes. Here, by investigating LysoTracker and cathepsin D distribution and by analyzing lysosome morphology using EM, we establish that ESCRT-I depletion leads to the enlargement of lysosomes. Importantly, our study suggests that the enlargement of lysosomes could be because of impaired degradation of lysosomal membrane proteins, including MCOLN1. We not only confirm that MCOLN1 is degraded from the lysosomal surface as recently reported (Lee et al, 2020) but also unravel that this degradation is mediated by ESCRT-I.

Although enlargement of lysosomes has been linked to their dysfunction or damage (Lakpa et al, 2021), we did not observe signs of abnormal composition of these organelles or loss of their integrity. However, we and others (Filimonenko et al, 2007; Maminska et al, 2016) previously found that endosomes or autophagosomes in ESCRT-I–deficient cells accumulate nondegraded cargo that is not properly targeted to lysosomes. Here, we show that it also concerns nondegraded lysosomal membrane proteins. Hence, we propose that although these enlarged lysosomes seem to retain their

degradative potential, the cargo destined for lysosomal degradation does not reach the lysosomal lumen. As a most likely consequence, in the absence of ESCRT-I, cells receive less nutrients (such as amino acids and cholesterol) from the lysosomal compartment. This reasoning explains the transcriptional activation of cholesterol biogenesis and RagC-mediated TFEB/TFE3 signaling in cells lacking ESCRT-I. Therefore, we propose that ESCRT-I deficiency leads to partial starvation due to shortage of lysosome-derived nutrients, that is, lysosomal nutrient starvation.

As reported, depletion of some components of ESCRT complexes, namely Hrs (ESCRT-0 subunit) or Vps4A/B, causes accumulation of cholesterol on LAMP1-positive structures (Du et al, 2012; Du et al, 2013). These studies, based on experiments performed in HeLa cells, concluded that Tsg101 or ESCRT-III subunits are not involved in cholesterol transport from the endolysosomal pathway, arguing that the roles of Hrs or Vps4A/B in this process are independent of other ESCRTs. However, our finding that the disruption of ESCRT-I complex by depletion of Tsg101 or Vps28 leads to accumulation of cholesterol in LAMP1-positive structures of RKO cells argues that proper endolysosomal transport of cholesterol requires a functional ESCRT machinery. The discrepancy between our results and published data (Du et al, 2012) regarding the effect of Tsg101 depletion may be because of cell type-specific effects.

The MCOLN1 protein has been shown to promote nuclear translocation of transcription factor TFEB, through activation of calcineurin phosphatase by Ca$^{2+}$ released from lysosomes (Medina et al, 2015). Calcineurin-dependent activation of TFE3 was also reported (Martina et al, 2016). We discovered that in cells lacking ESCRT-I, the nuclear translocation of TFEB and TFE3 was associated with an elevated abundance of nondegraded MCOLN1 on enlarged lysosomes. Thus, we initially hypothesized that accumulation of MCOLN1 may serve as a mechanism to activate these transcription factors by promoting their calcineurin-dependent dephosphorylation. However, although we observed that upon ESCRT-I depletion, activation of TFEB and TFE3 required Ca$^{2+}$-dependent signaling, inhibition of this signaling did not prevent TFEB dephosphorylation. Hence, no further increase in nuclear abundance of these factors in ESCRT-I–deficient cells upon Ca$^{2+}$ signaling inhibition must be due to mechanisms other than calcineurin-mediated dephosphorylation that remains to be addressed. A potential underlying mechanism may involve regulation of nucleo-cytoplasmic shuttling of transcription factors that is mediated by calcium ions (Sweitzer & Hanover, 1996; Holaska et al, 2002). In yeast, ESCRT dysfunction causes intracellular Ca$^{2+}$ accumulation resulting in hyperactivation of calcineurin (Zhao et al, 2013a; Zhao et al, 2013b; Schmidt et al, 2020). Although our

showing intracellular distribution of TFEB (green) with respect to LAMP1 protein (red) or DAPI-stained nuclei (blue) in ESCRT-I–depleted (siTsg101#2) or control (siCtrl#1) cells upon 2 h treatment with vehicle (Vh, DMSO) or 1 μM INK128. Scale bar, 100 μm. **(D, E)** Dot plots showing the percentage of cells with nuclear TFEB localization (D) or TFEB colocalization with LAMP1-positive structures (E) in ESCRT-I–depleted (siTsg101#2) or control (siCtrl#1) cells. Values derived from independent experiments (dots) and their means (n = 4 ± SEM) are presented. ns—nonsignificant, *P < 0.05, **P < 0.01, ****P < 0.0001. **(F)** Single-plane confocal images of fixed RKO cells at 48 hpt showing intracellular distribution of TFEB or TFE3 (green) with respect to DAPI-stained nuclei (blue) in control or ESCRT-I–depleted cells expressing or not wild-type (WT) or constitutively active (S75L) HA-GST-RagC protein (red). Scale bar, 20 μm. Dot plots on the right showing the percentage of cells with nuclear TFEB or TFE3 localization. Values derived from independent experiments (dots) and their means (n = 3 ± SEM) are presented. Statistical significance tested by comparison to siCtrl#1 conditions. ns—nonsignificant (P ≥ 0.05), *P < 0.05, **P < 0.01. **(G)** Western blots showing levels of phosphorylation of the indicated proteins as well as total levels of TFEB, Tsg101, and HA-GST-RagC in ESCRT-I–depleted (siTsg101#2) or control (siCtrl#1) cells at 48 hpt with ectopic expression of the HA-GST-RagC protein (WT or S75L). Vinculin was used as a loading control. The bottom graph shows phosphorylation levels of Ser122 of TFEB measured by densitometry analysis of Western blotting bands, including those shown above. Values derived from independent experiments (dots) and their means (n = 3 ± SEM) are presented. Statistical significance tested by comparison to siCtrl#1. ns, nonsignificant, *P < 0.05.

study points to some involvement of $Ca^{2+}$ in signaling activated upon ESCRT-I deficiency in mammalian cells, further investigation is required to address whether it leads to calcineurin activation.

Aberrant cholesterol transport from lysosomes has been associated with increased cytoplasmic $Ca^{2+}$ levels (Tiscione et al, 2019) and MiT-TFE activation (Willett et al, 2017; Boutry et al, 2019; Contreras et al, 2020). Although we observed a similar association in cells lacking ESCRT-I, we could not confirm a causal relationship between improper intracellular cholesterol distribution and activation of TFEB/TFE3 signaling. On the other hand, availability of lysosomal cholesterol was shown to stimulate mTORC1 signaling, resulting in increased S6K and 4E-BP1 phosphorylation (Castellano et al, 2017; Lim et al, 2019; Davis et al, 2021). We observed that upon ESCRT-I deficiency, the impaired delivery of cholesterol via endolysosomal trafficking is not associated with general inhibition of mTORC1 signaling. However, it is possible that, despite the shortage of cholesterol derived from endolysosomal trafficking in cells lacking ESCRT-I, the general mTORC1 signaling is maintained by additional intracellular stimuli. They could involve elevated cholesterol biosynthesis reported here or induction of growth factor signaling cascades that have been shown to occur upon ESCRT dysfunction (Szymanska et al, 2018).

Cancer cells often exhibit constitutive activation of MiT-TFE transcription factors that improves lysosomal degradation to support accelerated rates of growth (Perera et al, 2019). Intriguingly, our data show that basal TFEB/TFE3 nuclear accumulation in RKO cells is not because of $Ca^{2+}$-MCOLN1-calcineurin signaling but depends on the presence of lipids in the medium. Our discovery that inhibition of $Ca^{2+}$ signaling in control cells increased TFEB/TFE3 nuclear abundance and reduced TFEB phosphorylation was consistent with a report showing that MCOLN1-$Ca^{2+}$ signaling stimulates mTOR kinase activity (Li et al, 2016).

Overall, our work unravels the complexity of regulating the TFEB/TFE3 pathway under basal conditions and upon starvation-like signaling due to inhibited delivery of cargo for lysosomal degradation. Understanding such interplay may point to new strategies for cancer treatment (Perera et al, 2019). Further studies may address whether cancer cells with increased basal lysosomal function via MiT-TFE signaling could be particularly dependent on lysosomal membrane protein turnover that, as we show here, is mediated by ESCRT-I.

# Materials and Methods

### Antibodies

The following antibodies were used: anti-Tsg101 (Cat. no. ab83) and anti-Vps28 (Cat. no. ab167172) from Abcam; anti-TFEB (Cat. no. 4240S), anti-P-TFEB (S122, Cat. no. 86843 and S211, Cat. no. 37681), anti-TFE3 (Cat. no. 14779S), anti-P-Ulk1 (S757, Cat. no. 6888), anti-P-4E-BP1 (Thr37/46, Cat. no. 2855), anti-P-S6K1 (Thr389, Cat. no. 9234), anti-mTOR (Cat. no. 2983), and anti-HA-Tag (Cat. no. 2367) from Cell Signaling Technologies; anti-LAMP1 (Cat. no. H4A3) from DSHB; anti-mono- and -polyubiquitinylated conjugates (Cat. no. BML-PW8810) from Enzo Life Sciences; anti-GFP (Cat. no. AF4240) from R&D

Systems; anti-GAPDH (Cat. no. sc-25778) and anti-lamin A/C (Cat. no. sc-7292) from Santa Cruz; anti-histone H3 (Cat. no. H0164), anti-LAMP1 (Cat. no. L-1418), anti-vinculin (Cat. no. V9131-.2ML), and anti-tubulin (Cat. no. T5168) from Sigma-Aldrich; secondary HRP-conjugated goat anti-mouse, goat anti-rabbit, and bovine anti-goat antibodies from Jackson ImmunoResearch; and secondary Alexa Fluor 488–conjugated donkey anti-mouse and Alexa Fluor 647–conjugated donkey anti-rabbit antibodies from Thermo Fisher Scientific.

### Plasmids

To obtain EGFP-MCOLN1 construct for lentiviral transduction, human MCOLN1 was amplified by PCR using the following oligonucleotides with overhangs (underlined): 5′-GACACCGACTCTAGAATGGTGAGC-AAGGGCGAGGAGC-3′ (forward) and 5′-AACTAGTCCGGATCCT-CAATTCACCAGCAGCGAATGC-3′ (reverse) from the mucolipin1-pEGFP C3 (plasmid #62960; Addgene) construct and subcloned into XbaI and BamHI restriction sites of the pLenti-CMV-MCS-GFP-SV-puro (plasmid #73582; Addgene) vector using the sequence- and ligation-independent cloning (SLIC) method as described elsewhere (Jeong et al, 2012). The mucolipin1-pEGFP C3 (plasmid #62960; Addgene) construct was a gift from Paul Luzio (Pryor et al, 2006). pRK5-HA GST RagC WT (plasmid #19304; Addgene) and pRK5-HA GST RagC 75L (plasmid #19305; Addgene) were a gift from David Sabatini (Sancak et al, 2008). psPAX2 (plasmid #12260; Addgene) and pMD2.G (plasmid #12259; Addgene) lentiviral packaging plasmids were a gift from Didier Trono.

### Cell culture and treatment

DLD-1 colon adenocarcinoma and HEK293 and HEK293T embryonic kidney cells were maintained in DMEM (M2279; Sigma-Aldrich) supplemented with 10% (vol/vol) FBS (F7524; Sigma-Aldrich) and 2 mM L-glutamine (G7513; Sigma-Aldrich). RKO colon carcinoma cells were cultured in Eagle's minimum essential medium (EMEM, ATCC, 30-2003) supplemented with 10% (vol/vol) FBS. Both cell lines were regularly tested as mycoplasma-negative, and their identities were confirmed by short tandem repeat profiling performed by the ATCC Cell Authentication Service.

Cycloheximide (01810; Sigma-Aldrich) was applied at 100 $\mu$g/ml concentration for 4 or 8 h. BAPTA-AM (sc-202488; Santa Cruz) was applied to chelate the intracellular pool of calcium at 10 $\mu$M concentration for 2 h. To inhibit MCOLN1 calcium channel or calcineurin activity, ML-SI1 (GW 405833; Cayman Chemical) or cyclosporine A (sc-3503; Santa Cruz) were used, respectively, for 2 h both at 25 $\mu$M. Bafilomycin A1 (B1793; Sigma-Aldrich) at 50 nM concentration was applied for 18 h to inhibit lysosomal degradation. To inhibit mTOR activity, INK128 (S2811; Selleckchem) was used at 1 $\mu$M concentration for 2 h. A total of 20 $\mu$M water-soluble cholesterol (C4951; Sigma-Aldrich) was added for 48 h. Cells were live-stained with 50 nM or 500 nM LysoTracker Red DND-99 (L7528; Thermo Fisher Scientific) for 30 min for live or fixed cell imaging, respectively. EMEM medium supplemented with delipidated FBS (S181L; Biowest) was used for 40 h to deprive cells of exogenous lipids. To culture cells in nutrient-deficient medium, Earle's balanced salts solution (EBSS, E2888; Sigma-Aldrich) was used.

## Cell transfection and lentiviral transduction

RKO cells were seeded on six-well plates ($0.8 \times 10^5$ cells/well) for Western blotting and quantitative real-time PCR (qRT-PCR) experiments, on P100 dish ($0.9 \times 10^6$ cells per dish) for cellular fractionation experiment or on 0.2% gelatin (G1890; Sigma-Aldrich)–covered 96-well plate (655-090; Grainer Bio-One) (0.25 or $0.4 \times 10^3$ cells/well) for confocal microscopy. DLD-1 cells were seeded on a 6-well plate ($0.8 \times 10^5$ cells/well) for Western blotting or on a 96-well plate (655-090; Grainer Bio-One) ($0.2 \times 10^3$ cells/well) for microscopy. Then 24 h after seeding, cells were transfected with 30 nM siRNAs using Lipofectamine RNAiMAX Transfection Reagent (13778150; Thermo Fisher Scientific) according the manufacturer's instructions and imaged or harvested after 48 or 72 h post transfection (hpt). The following Ambion Silencer Select siRNAs (Thermo Fisher Scientific) were used: Negative Control No. 1 (siCtrl#1, 4390843) and Negative Control No. 2 (siCtrl#2, 4390846); siTsg101#1 (s14439), siTsg101#2 (s14440), siVps28#1 (s27577), siVps28#2 (s27579), siTFEB (s15496), and siTFE3 (s14031). In experiments with simultaneous knockdown of three genes, the total concentration of siRNA was adjusted to 60 nM using siCtrl#1. For overexpression of HA-GST-RagC constructs, 24 h after siRNA transfection, RKO cells were transfected with plasmids using Lipofectamine 2000 Transfection Reagent (11668019; Thermo Fisher Scientific).

For overexpression of EGFP-tagged MCOLN1, lentiviral particles were produced in HEK293T cells using packaging plasmids: psPAX2 and pMD2.G, as described elsewhere (Barde et al, 2010). Subsequently, $1 \times 10^6$ RKO cells were grown in 5 ml of virus-containing EMEM medium on P60 dish for 24 h. Then, cells were split and grown in selection EMEM medium containing 1 µg/ml puromycin for 72 h.

For CRISPR/Cas9–mediated knockout of the *TSG101* gene, two control nontargeting and four targeting different gRNA sequences (Doench et al, 2016) were cloned into the lentiCRISPRv2 vector and were lentivirally introduced into RKO cells in the same way as described above for EGFP-MCOLN1 overexpression. The efficiency of gene expression silencing was tested by Western blotting, and two targeting sequences causing the strongest reduction of Tsg101 protein levels were chosen for further experiments. The sequences of gRNAs used in this study are: gCtrl#1- CGCTTCCGCGGCCCGTTCAA, gCtrl#2- CTGAAAAAGGAAGGAGTTGA, gTsg101#1- AGGGAACTAAT-GAACCTCAC, gTsg101#2- ATCCGCCATACCAGGCAACG. For analysis of the effects of *TSG101* knockout, RKO cells were transduced with lentiviral particles containing control or gene-targeting vectors for 1 h and selected with 1 µg/ml puromycin (58-58-2; TOKU-E) for 72 h.

## Lysosomal staining

For live cell imaging, cells were incubated for 30 min with 50 nM LysoTracker as described elsewhere (Hirst et al, 2015). Subsequently, cells were washed with probe-free medium and immediately imaged. For fixed cell imaging, cells were incubated for 30 min with 500 nM LysoTracker and fixed with 3.6% paraformaldehyde for 15 min on ice followed by 15 min incubation at room temperature. After three washes with PBS, cells were immunostained.

## Immunofluorescence staining and microscopy

Cells seeded on 0.2% gelatin–coated plates (655-090; Greiner Bio-One) were fixed with 3.6% paraformaldehyde at room temperature

and immunostained as described elsewhere (Maminska et al, 2016). Cell nuclei were marked with DAPI (D9542; Sigma-Aldrich), Hoechst (H1399; Thermo Fisher Scientific), or DRAQ7 (D15106; Thermo Fisher Scientific) dye, as indicated in the figure legends. Also, 1 µg/ml pepstatin A BODIPY FL Conjugate (P12271; Thermo Fisher Scientific) was used to probe cathepsin D in fixed cells as described (Chen et al, 2000). Filipin III (F4767; Sigma-Aldrich) was used to stain intracellular cholesterol according to manufacturer instructions. Plates were scanned using Opera Phenix high content screening microscope (PerkinElmer) with 40 × 1.1 NA water immersion objective. Harmony 4.9 software (PerkinElmer) was applied for image acquisition and their quantitative analysis. For quantification of chosen parameters (mean area of vesicular structures, fluorescence intensity per structure, percentage of cells with nuclear staining, or colocalization expressed as Pearson's correlation coefficient), more than 10 microscopic fields were analyzed per each experimental condition. Maximum intensity projection images were obtained from three to five z-stack planes with 1-µm interval. Pictures were assembled in ImageJ and Photoshop (Adobe) with only linear adjustments of contrast and brightness.

## Western blotting

Cells were lysed in RIPA buffer (1% Triton X-100, 0.5% sodium deoxycholate, 0.1% SDS, 50 mM Tris pH 7.4, 150 mM NaCl, and 0.5 mM EDTA) supplemented with protease inhibitor cocktail (6 µg/ml chymostatin, 0.5 µg/ml leupeptin, 10 µg/ml antipain, 2 µg/ml aprotinin, 0.7 µg/ml pepstatin A, and 10 µg/ml 4-amidinophenylmethanesulfonyl fluoride hydrochloride; Sigma-Aldrich) and phosphatase inhibitor cocktails 2 and 3 (P5726 and P0044; Sigma-Aldrich). Protein concentration was measured with the BCA Protein Assay Kit (23225; Thermo Fisher Scientific). Subsequently, 15–25 µg of total protein per sample were resolved on 8–14% SDS–PAGE and transferred onto a nitrocellulose membrane (Amersham Hybond, 10600002; GE Healthcare Life Science). Membranes were blocked in 5% milk in PBS followed by incubation with specific primary and secondary antibodies. For signal detection, the Clarity Western ECL Substrate (170-5061; Bio-Rad) and ChemiDoc imaging system (Bio-Rad) were applied. Densitometric analysis of Western blotting bands was performed using Image Lab 6.0.1 software (Bio-Rad). The raw data were normalized to vinculin band intensities and presented as fold levels to the average of siCtrl#1 and siCtrl#2.

## Cell fractionation

Cellular fractionation was performed as described elsewhere (Suzuki et al, 2010). Briefly, cells growing on the P100 dish were washed with ice-cold PBS, scraped, and collected in 1.5-ml microcentrifuge tube. After centrifugation (10 s, $1.7 \times 10^3$ g), the pellet was resuspended in 900 µl of ice-cold 0.1% NP40 (IGEPAL CA-630, I8896; Sigma-Aldrich) in PBS, and 300 µl of the lysate (whole cell lysate fraction, W) was transferred to a separate tube. The remaining material was centrifuged (10 s, $1.2 \times 10^4$ g), and the pellet was resuspended in 1 ml of ice-cold 0.1% NP40 in PBS and centrifuged (10 s, $1.2 \times 10^4$ g). The pellet (~20 µl) was resuspended in 180 µl of 1 × Laemmli sample buffer (nuclear fraction, N). Lysates were sonicated and boiled for 1 min at 95°C.

## Quantitative real-time PCR (qRT-PCR)

Total RNA was isolated from cells with the High Pure Isolation Kit (11828665001; Roche) according to the manufacturer's instruction. For cDNA preparation, 500 ng of total RNA, random nonamers (R7647; Sigma-Aldrich), oligo(dT)23 (O4387; Sigma-Aldrich), and M-MLV reverse transcriptase (M1302; Sigma-Aldrich) were used. Primers were designed using the NCBI Primer designing tool and custom-synthesized by Sigma-Aldrich. The sequences of primers were listed in Supplementary Table. For two to three technical repeats for each experimental conditions, cDNA sample amplification was performed with the KAPA SYBR FAST qPCR Kit (KK4618; KapaBiosystems) using the 7900HT Fast Real-Time PCR thermocycler (Applied Biosystems). Obtained data were normalized according to the expression level of the *GAPDH* (glyceraldehyde 3-phosphate dehydrogenase) housekeeping gene. Results are presented as fold change compared to siCtrl#1.

## Transcriptome analysis by RNA sequencing (RNA-Seq)

Transcriptome analysis of RKO cells was performed as described elsewhere (Kolmus et al, 2021). Briefly, the cell pellet was collected 72 h posttransfection with siRNAs. To generate sequencing library, Ion AmpliSeq Transcriptome Human Gene Expression Panel (Thermo Fisher Scientific) was used. Sequencing was performed with the Ion PI Hi-Q Sequencing 200 Kit (Thermo Fisher Scientific) using the Ion Proton instrument. Alignment of reads to the hg19 AmpliSeq Transcriptome ERCC v1 was performed with the Torrent Mapping Alignment Program (version 5.0.4; Thermo Fisher Scientific), followed by transcript quantification with HTseq-count (version 0.6.0). Differential gene expression analysis was performed for genes with more than 100 counts across conditions using the R package DESeq2 (version 1.18.1; [Love et al, 2014]). Nonprotein coding genes were excluded from the analysis. The expression levels in all samples were normalized to those in siCtrl#1-transfected cells. Only genes characterized by adjusted *P*-value < 0.05 were considered as significant. Obtained counts were transformed using the transcript per million normalization method and converted to obtain Z-scores. The set of differentially expressed genes that were common for all on-target siRNAs was subjected to gene ontology analysis of biological processes using clusterProfiler (version 3.6.0; [Yu et al, 2012]) and corrected for multiple testing using the Benjamini–Hochberg method. To reduce the redundancy of terms, a 0.6 cutoff was applied. Heatmaps of differentially expressed genes were generated using Complex-Heatmap (version 1.17.1; [Gu et al, 2016]). The abovementioned calculations and visualizations were performed in R version 3.4.4 (https://www.R-project.org).

## EM sample preparation, processing, and imaging

RKO cells were seeded on Nunc Thermanox coverslips (150067; Thermo Fisher Scientific) on a 24-well plate. Furthermore, 24 h after seeding, cells were transfected with 30 nM siRNAs. Then 72 h after transfection, cells were fixed in 2.5% glutaraldehyde for 2 h, then washed three times with PBS, postfixed with 1% osmium tetroxide for 1 h, washed with water, and incubated in 1% aqueous uranyl acetate overnight at 4°C. Then the cells were dehydrated with increasing dilutions of ethanol, infiltrated with epoxy resin (45-359-1EA-F; Sigma-Aldrich), embedded using BEEM capsules (Hanson et al, 2010), and incubated at 60°C for 72 h. Then polymerized blocks were trimmed and cut with a Leica ultramicrotome (EM UC7) for ultrathin sections (65 nm thick) and collected on copper grids, mesh 300 (AGG2300C; Agar Scientific). Specimen grids were examined with a transmission electron microscope Tecnai T12 Bio-Twin (FEI) equipped with a 16-megapixel TemCam-F416 (R) camera (TVIPS GmbH).

## Statistical analysis

Data are shown as mean ± SEM from at least three independent biological experiments. Statistical analysis was performed using the Prism 8.4.3 (GraphPad Software) using the unpaired two-tailed *t* test (for qRT-PCR analysis, Western blotting densitometry and % of cells with TFEB or TFE3 in the nucleus from confocal microcopy analysis) or paired two-tailed *t* test (for quantified parameters from confocal microcopy analysis representing fluorescence intensity, mean structure area, and Pearson's correlation coefficient). The significance of mean comparison is annotated as follows: ns, nonsignificant ($P \geq 0.05$) or indicated with exact *P*-value, $*P < 0.05$, $**P < 0.01$, $***P < 0.001$, $****P < 0.0001$. Results were considered significant when $P < 0.05$.

# Data Availability

The RNA sequencing data have been deposited to Gene Expression Omnibus under the accession number: GSE178665 (https://www.ncbi.nlm.nih.gov/geo/query/acc.cgi?acc=GSE178665).

# Supplementary Information

# Acknowledgments

We are grateful to Jacek Jaworski for providing reagents. We also thank Kamil Jastrzębski, Agata Poświata, Lidia Wolińska-Nizioł, Daria Zdżalik-Bielecka, and Jacek Jaworski for critical reading of the manuscript. The work was supported by the TEAM grant (POIR.04.04.00-00-20CE/16–00) to M Miaczynska and by the HOMING grant (POIR.04.04.00-00-1C54/16-00) to J Cendrowski, both from the Foundation for Polish Science co-financed by the European Union under the European Regional Development Fund. E Szymańska and M Grębowicz-Maciukiewicz were supported by the Opus grant (2020/37/B/NZ3/02991) from the National Science Centre to E Szymańska.

## Author Contributions

M Wrobel: conceptualization, data curation, formal analysis, validation, investigation, visualization, methodology, and writing—original draft, and this author and J Cendrowski contributed equally.

J Cendrowski: conceptualization, formal analysis, supervision, funding acquisition, validation, investigation, visualization, project administration, writing—original draft, review, and editing, and this author and M Wrobel contributed equally.
E Szymańska: validation, investigation, and visualization.
M Grębowicz-Maciukiewicz: investigation and visualization.
N Budick-Harmelin: investigation.
M Macias: visualization and methodology.
A Szybinska: methodology.
M Mazur: investigation.
K Kolmus: formal analysis.
K Goryca: formal analysis.
M Dąbrowska: methodology.
A Paziewska: methodology.
M Mikula: resources and data curation.
M Miaczynska: conceptualization, resources, supervision, funding acquisition, and writing—original draft, review, and editing.

## Conflict of Interest Statement

The authors declare that they have no conflict of interest.

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
