## [Reviewer comments · Life Science Alliance]

Life Science Alliance

ESCRT-I fuels lysosomal degradation to restrict TFEB/TFE3 signaling via the Rag-mTORC1 pathway

Marta Wróbel, Jarosław Cendrowski, Ewelina Szymańska, Malwina Grębowicz-Maciukiewicz, Noga Budick-Harmelin, Matylda Macias, Aleksandra Szybinska, Michał Mazur, Krzysztof Kolmus, Krzysztof Goryca, Michalina Dąbrowska, Agnieszka Paziewska, Michał Mikula, and Marta Miaczynska

DOI: <https://doi.org/10.26508/lsa.202101239>

Corresponding author(s): Marta Miaczynska, International Institute of Molecular and Cell Biology and Jarosław Cendrowski, International Institute of Molecular and Cell Biology

Review Timeline:

Submission Date:	2021-09-21
Editorial Decision:	2021-09-21
Revision Received:	2022-02-25
Editorial Decision:	2022-03-09
Revision Received:	2022-03-14
Accepted:	2022-03-15

Transaction Report:

Please note that the manuscript was previously reviewed at another journal and the reports were taken into account in inviting a revision for publication at Life Science Alliance prior to submission to Life Science Alliance.

Referee #1 Review

Report for Author:

The manuscript reports that loss of ESCRT-I results in severe lysosomal defects, including defects in size regulation, decrease in lysosomal catabolic activity, protein and cholesterol accumulation. Loss of ESCRT function leads to an increase in Ca²⁺, and presumably activates the phosphatase Calcineurin, which in turn dephosphorylates TFEB/TFE3, two transcription factors. This leads to the translocation of these transcription factors from lysosomes into the nucleus where they up-regulated many genes, including genes required to boost lysosome biogenesis, catabolism and autophagy. Responsible for the increase in Ca²⁺ signaling is the accumulation of the Ca²⁺ channel MCOLN1 on lysosomes.

Hence loss of ESCRT function triggers a Ca²⁺ dependent stress to adapt gene expression with TFEB/TFE3 in an attempt to restore lysosome biogenesis.

These are very interesting findings. Two major points should be substantiated prior to publication:

1. The authors suggest that MCOLN1 is degraded in an ESCRT dependent manner. The data show that MCOLN1 is up-regulated ESCRT mutants (also transcriptionally) and accumulates on lysosomes, but also on other structures that are neither Lamp1 positive nor lysotracker positive. (a) What are these other structures? (b) No evidence is provided that indeed the turnover of MCOLN1 is ESCRT dependent and blocked in ESCRT-I mutants (e.g. using CHX chase experiments +/- Baf should be fairly easy) (c) Data is lacking that MCOLN1 is sorted directly from the lysosomal membrane into lumen (this might be hard to convincingly demonstrate - but the authors could just tune down that specific statement. It will not affect the overall conclusion of the study).
2. Loss of ESCRT-I will disrupt several cellular processes. How can the authors be sure that the TFEB/TFE3 activation is specifically caused by lysosomal defects and Ca²⁺ efflux via accumulating MCOLN1? While the authors use ML-SI1 to inhibit MCOLN1, the data is difficult to interpret: Inhibition of MCOLN1 leads to a doubling of nuclear TFEB localization also in the

control cells. The number of control cells and tsg101 KD cells with nuclear TFEB is similar (around 20%), but still twice as much as compared to untreated control cells (10%). Doesn't that imply that there are other sources of Ca²⁺ influx that activate TFEB in ESCRT mutants? Also, inhibition of Calcineurin could be tested. How does inhibition of MCOLN1 and/or Ca²⁺ signaling affect TFEB phosphorylation?

There are a few minor points:

As the author point out in the discussion: a recent paper showed that the ESCRT machinery can directly sort membrane proteins from the limiting membrane of lysosomes into the lumen for degradation (PMID: 34297722). Hence I think that it is no longer warranted to state in the abstract ' However, whether mammalian ESCRTs perform a similar function at lysosomes'. This holds also true for the introduction. Of note, this does not limit the impact of this paper.

Some phenotypes of ESCRT-I depleted cells are consistent with earlier reports that additionally reported the formation of defective endosomes and severe defects in the maturation of lysosomal protease such as Cathepsins and defects in Mannose 6-phosphate receptor trafficking (e.g.: PMID: 11208108, 22479596, 16014378).

It seems that elevated Ca²⁺ flux and calcineurin hyperactivation is a general feature of ESCRT mutants and appears to be evolutionary conserved also in yeasts (PMID: 23026396, 23933635, 32611771). Perhaps this could be included in the discussion. Who knows: maybe also archaea and other species use Ca²⁺ signaling to adapt to defects in ESCRT function.

Referee #2 Review

Report for Author:

The manuscript: " ESCRT-I controls lysosomal membrane protein homeostasis and restricts MCOLN1- dependent TFEB/TFE3 signaling" is a survey of some of the effects of depleting Tsg101, an ESCRT-I subunit which has been extensively characterized in previous studies in both yeast and animal cells. The manuscript documents some effects that contribute new aspects of understanding lysosomal function and how ESCRTs contribute to it. However, the scope of the discoveries seem limited and are likely not on the scale that readers of this journal would expect. Although the work touches on a number of aspects, at least one would need more in-depth analysis to make a significant mechanistic advance.

The authors show data to support the idea that ESCRT-I mediates the lysosomal degradation of late endosomal and lysosomal membrane proteins, and that it is lack of degradation that leads to large lysosomes. The main weakness of this work is that many of these observations have been made already about ESCRT-i, and it is not clear whether the incremental observations shown here add to the overall understanding a general reader of this journal would have of what ESCRT-I does.

For instance:

- The authors say that the discovered ESCRT-I limits the size of lysosomes. Although a better view is that loss of ESCRT--I swells late-endosomes/lysosomes, the main weakness of this aspect of the paper is that this observation has been made many times before. The authors explanation that their focus on using lysotracker rather than Lamp1 is more specific to lysosomes is incorrect, since both late endosomes and lysosomes are acidic and accumulate lysotracker and contain Lamp1.
- The idea that loss of ESCRT-I function impairs degradation of ubiquitinated membrane proteins is well established in the literature, as is the specific observation of ubiquitin accumulating on endosomal compartments when ESCRT-I is impaired.
- That there are transcriptome changes when ESCRT-I is altered is also documented. Indeed, this group has already shown such changes when ESCRT-I is compromised by the loss of other of its subunits. There is no comparison provided for how loss of TSG101 or Vps28 compares with loss of the Vps37* components. However, for the changes regarding cohorts of genes upon Tsg101 depletion, should match closely with changes in Vps37* loss if general conclusions about ESCRT-I are to be made. If these changes do match, they make only a minor advance since the transcriptome changes to the latter are already published.
- Changes in cholesterol in cells lacking full ESCRT function have been noted before, as this study does. There is a discrepancy in the current manuscript vs previous work as to whether loss of Tsg101 does disrupt cholesterol distribution (this study) or whether it does not (previous studies). This difference is not further explored or resolved, so its not clear whether this is important or not.
- The work shows that mucolipin-GFP expressed via transient transfection accumulates to higher levels in cells lacking ESCRT-I. Its not clear though, whether this because a portion of mucolipin is degraded by routing through the MVB pathway during its biogenesis, or, as hypothesized here, whether it is degraded by a newly described route whereby existing lysosomal membrane proteins are routed through an different ESCRT-driven sorting step. More work would be required to take these preliminary observations and detail the type of trafficking route that has been established in yeast and has been somewhat established in mammalian cells. A key thing missing is showing that mucolipin undergoes degradation in the lysosome normally and that this rate is slowed in Tsg101-depleted cells.

Referee #3 Review

Report for Author:

In this manuscript, the authors show that ESCRT-I restricts the size of lysosomes and promotes the degradation of proteins from lysosomal membranes such as LAMP1 and MCOLN1. Also, the lack of ESCRT-I promoted abnormal cholesterol accumulation in lysosomes and activated TFEB/TFE3 transcription factors in a calcium-dependent manner. Although some results are interesting, the message of the manuscript is not clear as well as the relevance of the findings. In this format, the paper is an accumulation of observations and the claims are not supported by the experimental evidence.

Major points

- 1- The claim that ESCRT-I restricts lysosomal size is not supported by the evidence. Firstly, this is a general claim that has been proved only in tumor cells of one category. Second, the authors use very few approaches parameters to conclude that ESCRT-1 restricts lysosomal size. Electron microscopy to proper identification of lysosomal morphology and determine the size. The claim is supported by loss of function approaches. If ESCRT-I restricts the lysosomal size, I will expect that the overexpression of these proteins reduces the size of lysosomes. Also, the gain of function approaches is required as well as depletion using CRISPR/Cas9 engineering of these proteins.
- The authors can not exclude that the depletion of ESCRT-I proteins is not globally affecting the endocytic compartment. The increasing vacuolization that they observed might be due to a global lysosomal dysfunction and not to a specific role of these complexes in lysosomal membrane protein degradation. In support of this concern, the authors described an accumulation of cholesterol and an elevation of lysotracker staining. In this regard is also not necessarily prove the claim that lysotracker staining increase means that the lysosome is functional. Elevation of lysotracker may also suggest a defect in acidification (more acidic in this case). Thus, a comprehensive characterization of the endocytic and lysosomal compartment is required upon depletion and overexpression of ESCRT-I proteins. electron microscopy is needed to identify a lysosomal storage disorder-like phenotype (accumulation of material inside vacuolized lysosomes=). Check the expression of soluble lysosomal proteins such as cathepsin D and B to exclude global accumulation instead of specific role on membrane proteins. Check lysosomal degradative capacity using BODIPY-PEPSTATIN or DQ-BSA.
- Another observation suggesting that the interpretation of data based on depletion of genes only may be wrong is the activation of the TFEB pathway. TFEB is activated under a myriad of stress conditions. In this regard, although the authors suggest that accumulation of lysosomal membrane protein is due to a block of their degradation upon ESCRT-I depletion, I suggest first excluding the possibility that such increases are not due to the transcriptional activation of these genes (MCOLN1 and LAMP1 are bonafide targets of TFEB).
- The authors showed that ESCRT-I deficiency leads to the prolonged activation of MiT-TFE signaling in a Ca²⁺- and MCOLN1-dependent manner. However, how ESCRT-I proteins modulate MCOLN1 activity and calcium release?
- The cholesterol part is attached to the manuscript without a clear flux. In addition, NPC1 that controls cholesterol efflux from the lysosome is upregulated upon ESCRT-I depletion. Thus, how do the authors reconcile this with an accumulation of lysosomal cholesterol? Zoncu's laboratory recently shown that lysosomal cholesterol accumulation activates mTORC1. Is mTOR active or inactive in this set-up. If active, how do the authors reconcile TFEB nuclear translocation in the presence of mTOR hyperactivity?

September 21, 2021

Re: Life Science Alliance manuscript #LSA-2021-01239-T

Prof. Marta Miaczynska
International Institute of Molecular and Cell Biology
Laboratory of Cell Biology
Ks. Trojdena 4
Warsaw 02-109
Poland

Dear Dr. Miaczynska,

Thank you for submitting your manuscript entitled "ESCRT-I controls lysosomal membrane protein homeostasis and restricts MCOLN1-dependent TFEB/TFE3 signaling" to Life Science Alliance. We invite you to re-submit the manuscript, revised according to your proposed revision plan. Please note that you do not need to provide gene changes between loss of TSG101 or Vps28 compared with loss of the Vps37* components, in response to Reviewer 2.

Thank you for this interesting contribution to Life Science Alliance. We are looking forward to receiving your revised manuscript.

Sincerely,

Eric Sawey, PhD
Executive Editor
Life Science Alliance
<http://www.lsa-journal.org>

B. MANUSCRIPT ORGANIZATION AND FORMATTING:

Wrobel et al

Rebuttal letter

General response from the authors

*We would like to thank the Referees for providing us with their constructive comments. Addressing them helped us to improve the scientific quality and raise the importance of our discoveries. The obtained new data reinforced most of our findings but also allowed drawing additional exciting conclusions. They are now reflected in the changed manuscript title, namely **"ESCRT-I fuels lysosomal degradation to restrict TFEB/TFE3 signaling via the Rag-mTORC1 pathway"**. The most important additions to the main message of the revised manuscript concern the mechanisms underlying the activation of TFEB/TFE3 signaling upon ESCRT-I depletion. In the original manuscript we focused on the involvement of MCOLN1-calcium-mediated signaling in TFEB/TFE3 activation. After extensive experimental revision, we came to the conclusion that although calcium-mediated signaling is required, it is not the primary causative factor for TFEB/TFE3 activation upon ESCRT-I depletion. We now conclusively demonstrate that ESCRT-I depletion activates TFEB/TFE3 signaling via the inhibition of a non-canonical mTORC1 pathway that involves Rag GTPase signaling. We propose that because ESCRT-I fuels lysosomal degradation by providing the lysosomes with cargo from multiple sources, i.e. endocytosis, autophagy and lysosomal membrane turnover, thus the lack of ESCRT-I leads to shortage of nutrients derived from lysosomal degradation. The resulting lysosomal nutrient starvation evokes transcriptional responses, including expression of genes responsible for cholesterol biosynthesis and autophagosome/lysosome biogenesis, the latter due to the inhibition of substrate-specific Rag GTPase-mTORC1 signaling.*

Referee #1:

The manuscript reports that loss of ESCRT-I results in severe lysosomal defects, including defects in size regulation, decrease in lysosomal catabolic activity, protein and cholesterol accumulation. Loss of ESCRT function leads to an increase in Ca²⁺, and presumably activates the phosphatase Calcineurin, which in turn dephosphorylates TFEB/TFE3, two transcription factors. This leads to the translocation of these transcription factors from lysosomes into the nucleus where they up-regulated many genes, including genes required to boost lysosome biogenesis, catabolism and autophagy. Responsible for the increase in Ca²⁺ signaling is the accumulation of the Ca²⁺ channel MCOLN1 on lysosomes.

Hence loss of ESCRT function triggers a Ca²⁺ dependent stress to adapt gene expression with TFEB/TFE3 in an attempt to restore lysosome biogenesis.

These are very interesting findings. Two major points should be substantiated prior to publication:

1. The authors suggest that MCOLN1 is degraded in an ESCRT dependent manner. The data show that MCOLN1 is up-regulated ESCRT mutants (also transcriptionally) and accumulates on lysosomes, but also on other structures that are neither Lamp1 positive nor lysotracker positive. (a) What are these other structures?

Authors: This is a very interesting point to address. MCOLN1 protein reaches lysosomal compartment via endocytosis and trafficking from the trans-Golgi Network (TGN) (PMID: 20074572). As we show in **Figure for the Referees**, we analyzed the abundance of GFP-MCOLN1 on the Golgi apparatus (marked with TGN46) or early endosomes (marked with EEA1), as compared to the abundance on LAMP1-positive late endosomes and lysosomes, in control or ESCRT-I-depleted RKO cells. We observed that in control cells, GFP-MCOLN1 is predominantly present on late endosomes and lysosomes. ESCRT-I depletion markedly increased GFP-MCOLN1 levels not only on these structures but also on early endosomes. The GFP-MCOLN1 abundance was also significantly elevated on the Golgi, however this increase was much weaker. We find these results very interesting, however we have shifted the focus of the revised manuscript away from the MCOLN1 accumulation (see our response to point 2). Therefore, for the sake of the clarity of our story we have decided not to include these data in the revised manuscript.

Figure for the Referees. (A-B) Single confocal plane images showing the distribution of stably overexpressed GFP-MCOLN1 with respect to endogenous markers of various intracellular compartments, LAMP1 and TGN46 (in A) or LAMP1 and EEA1 (in B) in fixed RKO cells transfected with control non-targeting (siCtrl#1 or #2) or ESCRT-I-targeting (siTsg101#2 or siVps28#1) siRNAs. Cell nuclei labelled with DAPI. Scale bars, 20 μ m. **(C)** Dot plots showing the fluorescence intensity of GFP-MCOLN1 (expressed in arbitrary units, a.u.) detected in the Golgi apparatus (TGN46-positive), early endosomes (EEA1-positive) or late endosomes/lysosomes (LAMP1-positive) from confocal microscopy images of control or ESCRT-I-depleted cells (including those shown in A and B). Values derived from independent experiments (dots) and their means ($n=4$ +/- SEM) are presented. Statistical significance tested by comparison to averaged values measured for siCtrl#1 and #2 (AveCtrl) using paired two-tailed Student t-test. ** $P<0.01$, *** $P<0.001$.

(b) No evidence is provided that indeed the turnover of MCOLN1 is ESCRT dependent and blocked in ESCRT-I mutants (e.g. using CHX chase experiments +/- Baf should be fairly easy)

Authors: We agree that this important piece of evidence was missing from the previous version. In the revised manuscript (new Fig. 2E), we show that although the levels of GFP-MCOLN1 are reduced upon CHX treatment in control cells, this reduction is not observed in the absence of Tsg101, confirming increased stability of the protein.

(c) Data is lacking that MCOLN1 is sorted directly from the lysosomal membrane into lumen (this might be hard to convincingly demonstrate - but the authors could just tune down that specific statement. It will not affect the overall conclusion of the study).

Authors: As indicated by the Referee, in the revised manuscript we do not conclude that MCOLN1 is sorted directly to the lysosomal lumen.

2. Loss of ESCRT-I will disrupt several cellular processes. How can the authors be sure that the TFEB/TFE3 activation is specifically caused by lysosomal defects and Ca^{2+} efflux via accumulating MCOLN1? While the authors use ML-SI1 to inhibit MCOLN1, the data is difficult to interpret: Inhibition of MCOLN1 leads to a doubling of nuclear TFEB localization also in the control cells. The number of control cells and tsg101 KD cells with nuclear TFEB is similar (around 20%), but still twice as much as compared to untreated control cells (10%). Doesn't that imply that there are other sources of Ca^{2+} influx that activate TFEB in ESCRT mutants? Also, inhibition of Calcineurin could be tested. How does inhibition of MCOLN1 and/or Ca^{2+} signaling affect TFEB phosphorylation?

Authors: We thank the Referee for this valuable observation. Addressing this comment showed that indeed the involvement of MCOLN1/ Ca^{2+} signaling in TFEB/TFE3 activation in ESCRT-I-depleted cells cannot be explained in a straightforward manner. As suggested, we included in our experiments the analysis of TFEB phosphorylation and we applied calcineurin inhibitor (cyclosporin, CsA). First of all, we observed strong reduction of TFEB phosphorylation at S112 in ESCRT-I-depleted cells (new Fig. 6B, 7A-B and S7A-B). Surprisingly, we discovered that treatment with ML-SI1, BAPTA or CsA did not restore this phosphorylation (new Fig. 6B). Moreover, we found that these compounds reduced TFEB phosphorylation in control cells, which explained the increased basal TFEB/TFE3 signaling that was commented by the Referee.

Eventually, we demonstrated that TFEB/TFE3 signaling upon ESCRT-I depletion is due to

the

inhibition of the Rag-dependent non-canonical mTORC1 pathway (new Fig. 7C-E and S7C-E). This is now the main message of the study.

There are a few minor points:

As the author point out in the discussion: a recent paper showed that the ESCRT machinery can directly sort membrane proteins from the limiting membrane of lysosomes into the lumen for degradation (PMID: 34297722). Hence I think that it is no longer warranted to state in the abstract ' However, whether mammalian ESCRTs perform a similar function at lysosomes'. This holds also true for the introduction. Of note, this does not limit the impact of this paper.

Authors: As indicated by the Referee we have removed this statement from the abstract. Moreover, we have referred to this study not only in the discussion but also in the introduction of the revised manuscript.

Some phenotypes of ESCRT-I depleted cells are consistent with earlier reports that additionally reported the formation of defective endosomes and severe defects in the maturation of lysosomal protease such as Cathepsins and defects in Mannose 6-phosphate receptor trafficking (e.g.: PMID: 11208108, 22479596, 16014378).

Authors: To address the functionality of lysosomes we stained cells with pepstatin A-BODIPY that detects cathepsin D. We detected high levels of cathepsin D in the enlarged LAMP1+ structures of ESCRT-I-depleted cells (new Fig. 1C). Moreover, by electron microscopy (EM) analysis, we observed enlarged lysosomes in these cells, in addition to enlarged endosomes and autophagosomes with undegraded cargo that were described before (new Fig. 1D and S3A-B). However, based on cathepsin D distribution and the lysosomal morphology observed by EM, we did not find symptoms of impaired degradative potential of the enlarged lysosomes. Based on these data and on the analysis of LysoTracker distribution we conclude that ESCRT-I deficiency does not lead to dysfunctional lysosomes. Instead, we propose that it leads to impaired delivery of cargo from endocytosis, autophagy and lysosomal membrane protein turnover for lysosomal degradation. This causes a shortage of lysosome-derived nutrients that activates signaling cues which resemble a cellular response to starvation.

It seems that elevated Ca²⁺ flux and calcineurin hyperactivation is a general feature of ESCRT mutants and appears to be evolutionary conserved also in yeasts (PMID: 23026396, 23933635, 32611771). Perhaps this could be included in the discussion. Who knows: maybe also archaea and other species use Ca²⁺ signaling to adapt to defects in ESCRT function.

Authors: Although we have toned down the importance of calcium signaling for TFEB/TFE3 activation upon ESCRT-I depletion in the revised manuscript, we refer to the studies indicated by the Referee in the discussion section of the revised manuscript.

Referee #2:

The manuscript: " ESCRT-I controls lysosomal membrane protein homeostasis and restricts

MCOLN1- dependent TFEB/TFE3 signaling" is a survey of some of the effects of depleting Tsg101, an ESCRT-I subunit which has been extensively characterized in previous studies in both yeast and animal cells. The manuscript documents some effects that contribute new aspects of understanding lysosomal function and how ESCRTs contribute to it. However, the scope of the discoveries seem limited and are likely not on the scale that readers of this journal would expect. Although the work touches on a number of aspects, at least one would need more in-depth analysis to make a significant mechanistic advance.

Authors: We agree that our study focuses on a narrow aspect of membrane trafficking, namely the functions of the ESCRT-I complex. Previous work by others and us (PMID: 15240819; 21757351; 24284069; 33419951) demonstrate that depletion of any of the core ESCRT-I components, such as Tsg101, leads to destabilization of the whole complex. Therefore, removal of Tsg101 serves as a model of ESCRT-I dysfunction. Nevertheless, to avoid unspecific findings, many of our results are based on depleting two ESCRT-I proteins Tsg101 or Vps28. Already in the previous version of the manuscript, by detailed and comprehensive analyses, we discovered several consequences of ESCRT-I depletion that have been overlooked by previous studies. We also provided mechanistic insights (e.g. TFEB/TFE3 involvement) into the regulation of transcriptional changes that occur upon ESCRT-I depletion. Therefore, we do not share the negative view of the Referee that this is just a survey of effects of depleting one protein.

As a result of an extensive experimental revision, we now show that ESCRT-I depletion evokes signaling responses to counteract the shortage of lysosome-derived nutrients. This concept, although drawn from studying the role of ESCRT-I, may be a valuable addition to the intensively studied subject of the crosstalk between endolysosomal trafficking and cell metabolism.

The authors show data to support the idea that ESCRT-I mediates the lysosomal degradation of late endosomal and lysosomal membrane proteins, and that it is lack of degradation that leads to large lysosomes. The main weakness of this work is that many of these observations have been made already about ESCRT-i, and it is not clear whether the incremental observations shown here add to the overall understanding a general reader of this journal would have of what ESCRT-I does.

Authors: In the previous manuscript version, we not only confirmed ESCRT-I-mediated degradation of lysosomal membrane proteins in mammalian cancer cells, that has been shown for vacuolar membrane proteins in yeast, but we also identified the cellular consequences of lack of this function. The main consequence discovered by us, the activation of the TFEB/TFE3 pathways, cannot be studied in yeast, hence we do not agree that our results are incremental.

In the revised version of the manuscript we provide another finding that has not been shown before as a consequence of ESCRT-I depletion, i.e. regulation of the substrate-specific Rag GTPase-dependent mTORC1 pathway (new Fig. 7).

For instance:

- The authors say that the discovered ESCRT-I limits the size of lysosomes. Although a better

view is that loss of ESCRT-I swells late-endosomes/lysosomes, the main weakness of this aspect of the paper is that this observation has been made many times before. The authors explanation that their focus on using lysotracker rather than Lamp1 is more specific to lysosomes is incorrect, since both late endosomes and lysosomes are acidic and accumulate lysotracker and contain Lamp1.

Authors: We do not agree that this part of our paper is a weakness. The observations made by other studies did not provide any explanation for the enlargement of lysosomes. This phenomenon could not be explained based on the previously proposed roles of ESCRT proteins (for instance in MVB formation).

We do not understand why the Referee criticizes use of LysoTracker that is a commonly used lysosomal marker (PMID: 23378628, 15051542, 26799652, 21285271, 31813797, 30314966 and many more). Late endosomes are indeed more acidic than endosomes but not to the same extent as lysosomes (PMID: 26448863, 23378628). The structures that we analyze, particularly upon ESCRT-I depletion, have the strongest LysoTracker staining, hence according to the text-book knowledge, we refer to them as lysosomes. Some LAMP1-positive structures may represent late endosomes that are not acidic enough to be detected by LysoTracker and we indeed observe such structures, visible in Fig. 2A and C, that are not enlarged. Hence, our strategy to study the lysosomal compartment in ESCRT-I-depleted cells is more accurate than performed in other studies.

- The idea that loss of ESCRT-I function impairs degradation of ubiquitinated membrane proteins is well established in the literature, as is the specific observation of ubiquitin accumulating on endosomal compartments when ESCRT-I is impaired.

Authors: We do not understand how this comment relates to our paper. We focus on novel findings related to the role of ESCRT-I in degradation of lysosomal membrane proteins and not on endosomal compartments.

- That there are transcriptome changes when ESCRT-I is altered is also documented. Indeed, this group has already shown such changes when ESCRT-I is compromised by the loss of other of its subunits. There is no comparison provided for how loss of TSG101 or Vps28 compares with loss of the Vps37* components. However, for the changes regarding cohorts of genes upon Tsg101 depletion, should match closely with changes in Vps37* loss if general conclusions about ESCRT-I are to be made. If these changes do match, they make only a minor advance since the transcriptome changes to the latter are already published.

Authors: The transcriptional consequences of depleting three isoforms of Vps37 that we published in the paper mentioned by the Referee (PMID: 33419951) were shown in a different cell line (DLD-1) than those of depleting Tsg101 or Vps28 that we show here (RKO cells). Although bioinformatic analysis performed in that paper did not indicate it, the transcriptional upregulation of TFEB/TFE3 targets and cholesterol biosynthesis genes occurs in DLD-1 cells lacking Vps37 isoforms, similarly as we show in RKO cells lacking Tsg101 or Vps28. We could easily include it in the manuscript, however this would be simply a repetition of results that will

make the manuscript longer and, as noted by the Referee, showing such comparison would make only a minor advance.

- Changes in cholesterol in cells lacking full ESCRT function have been noted before, as this study does. There is a discrepancy in the current manuscript vs previous work as to whether loss of Tsg101 does disrupt cholesterol distribution (this study) or whether it does not (previous studies). This difference is not further explored or resolved, so its not clear whether this is important or not.

Authors: We clearly show the involvement of Tsg101 (and another ESCRT-I component Vps28) in cholesterol trafficking, hence we provide strong evidence for the effect. The paper by Du X. et al (PMID: 22832105), to which the Referee refers, showed that Tsg101 was not involved, therefore reporting the absence of evidence, which is not the same as the evidence of absence. We not only show disrupted cholesterol trafficking but also an increased transcriptional response rescued by cholesterol supplementation. Resolving the difference between the previous report and our current work would be beyond the scope of our study.

- The work shows that mucolipin-GFP expressed via transient transfection accumulates to higher levels in cells lacking ESCRT-I. Its not clear though, whether this because a portion of mucolipin is degraded by routing through the MVB pathway during its biogenesis, or, as hypothesized here, whether it is degraded by a newly described route whereby existing lysosomal membrane proteins are routed through an different ESCRT-driven sorting step. More work would be required to take these preliminary observations and detail the type of trafficking route that has been established in yeast and has been somewhat established in mammalian cells. A key thing missing is showing that mucolipin undergoes degradation in the lysosome normally and that this rate is slowed in Tsg101-depleted cells.

Authors: The Referee understood our experimental settings incorrectly and overlooked important results that were already included in the previous version of the manuscript.

*First, we did not perform transient transfection. We established an RKO cell line, stably expressing GFP-MCOLN1. We did not observe an overload of the biosynthetic pathway by the stably expressed protein (see **Figure for the Referees** and the response to point 1 of Referee #1). We saw this protein localized primarily on the membranes of endosomes and lysosomes as described by Lee et al. that reported GFP-MCOLN1 to be a marker of lysosomal protein degradation (PMID: 32916093).*

Second, we did show this "key thing" that MCOLN1 undergoes degradation in lysosomes normally, in Fig. 2D of the original manuscript. However, we realized that the way we presented the data was suboptimal. So in the revised manuscript we present the data in a different way - one combined graph for vehicle- and BafA1-treated cells instead of two separate graphs.

Third, in the revised version we present the results of CHX chase experiments (new Fig. 2E) that the turnover of MCOLN1 is ESCRT-I-dependent.

Referee #3:

In this manuscript, the authors show that ESCRT-I restricts the size of lysosomes and promotes the degradation of proteins from lysosomal membranes such as LAMP1 and MCOLN1. Also, the lack of ESCRT-I promoted abnormal cholesterol accumulation in lysosomes and activated TFEB/TFE3 transcription factors in a calcium-dependent manner. Although some results are interesting, the message of the manuscript is not clear as well as the relevance of the findings. In this format, the paper is an accumulation of observations and the claims are not supported by the experimental evidence.

Major points

1- The claim that ESCRTI restricts lysosomal size is not supported by the evidence. Firstly, this is a general claim that has been proved only in tumor cells of one category. Second, the authors use very few approaches parameters to conclude that ESCRT-1 restricts lysosomal size. Electron microscopy to proper identification of lysosomal morphology and determine the size. The claim is supported by loss of function approaches. If ESCRT-I restricts the lysosomal size, I will expect that the overexpression of these proteins reduces the size of lysosomes. Also, the gain of function approaches is required as well as depletion using CRISPR/Cas9 engineering of these proteins.

Authors: As requested, we analyzed the morphology of endolysosomal compartments in wildtype and ESCRT-I-depleted cells by electron microscopy (new Fig.1D and S3A-B). These analyses confirmed the existence of enlarged lysosomes upon ESCRT-I depletion, as well as previously described alterations in other endolysosomal compartments such as MVBs and autophagosomes.

The request of the Referee about overexpressing the ESCRT-I proteins is not feasible and would not solve the questioned issue. First, it is not possible to effectively overproduce individual ESCRT-I components, as excess uncomplexed subunits are degraded (PMID 18077552 and our unpublished data). Second, ESCRT complexes mediate degradation of ubiquitylated membrane proteins and thereby restrict lysosomal size. Increasing ESCRT protein levels per se would not increase the amounts of membrane proteins to be degraded, hence would not reduce lysosomal size.

As requested by the Referee, in the revised manuscript, we show data based on CRISPR/Cas9-mediated depletion of Tsg101 (new Fig. S1C-D), which recapitulated the enlargement of lysosomes observed in siRNA experiments. We considered this as more important than showing enlarged lysosomes in another non-colon cancer cell line. We wish to stress that although RKO and DLD-1 cells are both derived from colon cancer, they differ in genetic background, shape and growth properties (PMID: 24042735). Still, they both exhibit lysosome enlargement upon ESCRT-I depletion. Moreover, we verified our key observation, namely activation of TFEB/TFE3 transcription factors, showing that it occurs not only in RKO cells but also in DLD-1 and HEK293 cell lines depleted of ESCRT-I (new Fig. S4A-B).

- The authors can not exclude that the depletion of ESCRT-I proteins is not globally affecting the endocytic compartment. The increasing vacuolization that they observed might be due to a global lysosomal dysfunction and not to a specific role of these complexes in lysosomal membrane protein degradation. In support of this concern, the authors described an accumulation of cholesterol and an elevation of lysotracker staining. In this regard is also not necessarily prove the claim that lysotracker staining increase means that the lysosome is functional. Elevation of lysotracker may also suggest a defect in acidification (more acidic in this case). Thus, a comprehensive characterization of the endocytic and lysosomal compartment is required upon depletion and overexpression of ESCRT-I proteins. electron microscopy is needed to identify a lysosomal storage disorder-like phenotype (accumulation of material inside vacuolized lysosomes). Check the expression of soluble lysosomal proteins such as cathepsin D and B to exclude global accumulation instead of specific role on membrane proteins. Check lysosomal degradative capacity using BODIPY-PEPSTATIN or DQ-BSA.

Authors: We thank the Referee for these insightful comments. To address these issues, we have performed electron microscopy (new Fig. 1D and S3A-B) and analyzed intracellular distribution of cathepsin D by staining fixed cells with pepstatin A-BODIPY (new Fig. 1C). We observed that the enlarged lysosomes contain high levels of cathepsin D and do not accumulate non-degraded cargo. We did not find signs of a lysosomal storage disorder-like phenotype. Based on these experiments, we conclude that the enlarged lysosomes upon ESCRT-I deficiency seem to retain their degradative potential and are not stressed. Instead, we propose that in cells lacking ESCRT-I, various types of cargo from endocytosis, autophagy and lysosomal membrane protein turnover do not reach lysosomes for degradation that causes a shortage of lysosome-derived nutrients.

According to our pepstatin A-BODIPY staining, the levels of cathepsin D increase in cells lacking ESCRT-I. We believe that this is caused by increased expression of a gene encoding this enzyme, as we observe in the transcriptomic data (not shown in the manuscript, but can be found in RNAseq data deposited in Gene Expression Omnibus (GEO), the accession code: GSE178665, the secure token: yvkzsseurpqhzqr), which is likely due to TFEB/TFE3 activation. We did not perform any uptake experiments to address degradation of endocytosed cargo as we know that ESCRT-I deficiency impairs endocytic uptake (our unpublished data) which would make the results of such experiments inconclusive.

- Another observation suggesting that the interpretation of data based on depletion of genes only may be wrong is the activation of the TFEB pathway. TFEB is activated under a myriad of stress conditions. In this regard, although the authors suggest that accumulation of lysosomal membrane protein is due to a block of their degradation upon ESCRT-I depletion, I suggest first excluding the possibility that such increases are not due to the transcriptional activation of these genes (MCOLN1 and LAMP1 are bonafide targets of TFEB).

Authors: We wish to point out that in our manuscript, we show protein levels of GFP-MCOLN1 that is ectopically expressed and therefore does not undergo transcriptional regulation

downstream of TFEB/TFE3. In addition, we show massive accumulation of ubiquitin on lysosomes. We see no premises to question the impaired degradation of lysosomal membrane proteins in ESCRT-I depleted cells that is in agreement with the recently published complementary study of Zhang et al (PMID 34297722).

- The authors showed that ESCRT-I deficiency leads to the prolonged activation of MiT-TFE signaling in a Ca²⁺- and MCOLN1-dependent manner. However, how ESCRT-I proteins modulate MCOLN1 activity and calcium release?

Authors: We do not claim that ESCRT-I regulates MCOLN1 activity or calcium release. Rather, we observe that the activation of calcium signaling is a consequence of ESCRT-I deficiency. Many previous studies have shown that impaired lysosomal homeostasis activates MCOLN1-dependent calcium signaling (PMID: 25720963, 32989250, 27357649). Moreover, as indicated by Referee #1, ESCRT mutants in yeast show elevated Ca²⁺ flux and calcineurin hyperactivation (PMID: 23026396, 23933635), which may be due to impaired plasma membrane remodeling that leads to influx of extracellular calcium (PMID: 32611771). Although our study points to some involvement of Ca²⁺ in signaling activated upon ESCRT-I deficiency in mammalian cells, we do not show elevated levels of Ca²⁺ or its potential source. Instead, urged by our new results regarding regulation of TFEB phosphorylation (new Fig. 6B, 7A-B and S7A; see also answer to point 2 of Referee #1), we have now refocused our study on the Rag-dependent non-canonical mTORC1 pathway as a mechanism underlying prolonged activation of MiT-TFE signaling upon ESCRT-I deficiency.

- The cholesterol part is attached to the manuscript without a clear flux. In addition, NPC1 that controls cholesterol efflux from the lysosome is upregulated upon ESCRT-I depletion. Thus, how do the authors reconcile this with an accumulation of lysosomal cholesterol? Zoncu's laboratory recently shown that lysosomal cholesterol accumulation activates mTORC1. Is mTOR active or inactive in this set-up. If active, how do the authors reconcile TFEB nuclear translocation in the presence of mTOR hyperactivity?

Authors: We thank the Referee for these comments. We have moved the part related to cholesterol trafficking earlier in the manuscript, which improved the overall flow of the presented story. In the revised manuscript we report that cholesterol is one of nutrients that is not properly transported via endolysosomal trafficking in cells lacking ESCRT-I. This leads to cholesterol starvation, which is counteracted by a homeostatic transcriptional response to synthesize new cholesterol. In fact, by applying a biochemical assay, we found that total cholesterol levels are not altered in these cells (our unpublished data). Hence, the shortage of endolysosomal cholesterol is counteracted by its biosynthesis, as it typically occurs in cells (PMID: 31848472).

As requested by the Referee, we measured the effect of ESCRT-I depletion on mTORC1 signaling. mTORC1 is a sensor of intracellular nutrients and studies from the Zoncu's laboratory show that cholesterol is one of the nutrients sensed by this kinase complex (PMID: 28336668, 33308480). Hence, we expected that the impaired delivery of cholesterol from the

endolysosomal system should rather inactivate mTORC1 signaling, which would be consistent with TFEB/TFE3 nuclear translocation. However, we found that phosphorylations of canonical mTORC1 targets were not reduced upon ESCRT-I depletion (new Fig. 7A-B and S7A-B). As we write in the discussion of the revised manuscript, lack of changes in general mTORC1 signaling could be due to intracellular events that counteract potential negative regulation due to cholesterol shortage. This could involve availability of cholesterol derived from its induced biosynthesis, or elevated calcium signaling that activates mTOR kinase (PMID: 27787197), or growth factor signaling cascades that are activated due to ESCRT-I depletion (PMID: 28797837).

Importantly, in the revised manuscript, we discover that, despite no changes in general mTORC1 signaling, lack of ESCRT-I leads to inactivation of the recently characterized substrate-specific Rag GTPase-mTORC1 pathway (PMID: 34253722, 32612235), that underlies activation of TFEB/TFE3 signaling (new Fig. 7C-G and S7C-E). Given that this pathway is regulated by lysosome-derived nutrients, we conclude that the impaired fueling of lysosomes with cargo destined for degradation due to ESCRT-I deficiency leads to insufficient delivery of nutrients derived from lysosomes, i.e. lysosomal nutrient starvation.

March 9, 2022

RE: Life Science Alliance Manuscript #LSA-2021-01239-TR

Prof. Marta Miaczynska
International Institute of Molecular and Cell Biology
Laboratory of Cell Biology
Ks. Trojdena 4
Warsaw 02-109
Poland

Dear Dr. Miaczynska,

Thank you for submitting your revised manuscript entitled "ESCRT-I fuels lysosomal degradation to restrict TFEB/TFE3 signaling via the Rag-mTORC1 pathway". We would be happy to publish your paper in Life Science Alliance pending final revisions necessary to meet our formatting guidelines. Please incorporate Reviewer 1's final comments.

-please add ORCID ID for secondary corresponding author - they should have received instructions on how to do so

FIGURE CHECKS:

-please confirm if the blots in Figure S2A are derived from a single, continuous gel

A. FINAL FILES:

B. MANUSCRIPT ORGANIZATION AND FORMATTING:

Sincerely,

Reviewer #1 (Comments to the Authors (Required)):

The authors have addressed most of my initial concerns. The paper shows convincingly that ESCRT mutant cells suffer from multiple defects in the endo-lysosomal compartment (including impaired MCOLN1 degradation) that are associated with defects in lysosomal mTOR signaling and subsequent TFEB/TFE3 nuclear translocation and the concomitant transcriptional changes. The defects in mTOR signaling result in part from a shortage in lysosomal amino acids (because fewer proteins are delivered for degradation), and hence impaired Rag dependent mTOR activation.

A minor point:

The section on Ca²⁺ signaling (p. 10) might profit from re-writing: this section was a bit difficult to understand for me e.g.:

'Surprisingly, BAPTA-AM or ML-SI1 treatment did not decrease the nuclear levels of TFEB and TFE3 in control cells, indicating that basal activation of MiT-TFE factors is not mediated by Ca²⁺ signaling (Fig.6A and S6). Intriguingly, these compounds had a tendency to increase the TFEB/TFE3 nuclear levels in control cells. However, Ca²⁺ chelation or inhibition of MCOLN1 or calcineurin prevented the nuclear accumulation of TFEB and TFE3 proteins due to Tsg101 depletion (Fig.6A and S6).

The above results showed that activation of MiT-TFE factors upon ESCRT-I deficiency requires Ca²⁺-dependent signaling.....'

I would argue that the data shows quite clearly that Ca²⁺ and Calcineurin in ESCRT mutants are involved in TFEB nuclear translocation. Yet at the end of this paragraph the authors conclude:

'Collectively, although we observed that calcium signaling is required for inducing TFEB/TFE3 transcription factors upon ESCRT-I depletion, the underlying mechanism of this regulation does not involve the canonical Ca²⁺-dependent dephosphorylation by calcineurin.'

Perhaps it would be better to argue: Collectively, we observed that calcium signaling is required for inducing TFEB/TFE3 nuclear translocation upon ESCRT-I depletion. Yet, the underlying mechanism of this regulation does not involve the canonical Ca²⁺-dependent dephosphorylation of TFEB/TFE3 by calcineurin.

March 15, 2022

RE: Life Science Alliance Manuscript #LSA-2021-01239-TRR

Prof. Marta Miaczynska
International Institute of Molecular and Cell Biology
Laboratory of Cell Biology
Ks. Trojdena 4
Warsaw 02-109
Poland

Dear Dr. Miaczynska,

Thank you for submitting your Research Article entitled "ESCRT-I fuels lysosomal degradation to restrict TFEB/TFE3 signaling via the Rag-mTORC1 pathway". It is a pleasure to let you know that your manuscript is now accepted for publication in Life Science Alliance. Congratulations on this interesting work.

DISTRIBUTION OF MATERIALS:

Again, congratulations on a very nice paper. I hope you found the review process to be constructive and are pleased with how the manuscript was handled editorially. We look forward to future exciting submissions from your lab.

Sincerely,
